# Infants' cortex undergoes microstructural growth coupled with myelination during development

Vaidehi S. Natu [1,9✉], Mona Rosenke[1,9], Hua Wu[2], Francesca R. Querdasi[1,3], Holly Kular[1], Nancy Lopez-Alvarez[1], Mareike Grotheer[1,4,5], Shai Berman[6], Aviv A. Mezer [6] & Kalanit Grill-Spector [1,7,8✉]

Development of cortical tissue during infancy is critical for the emergence of typical brain functions in cortex. However, how cortical microstructure develops during infancy remains unknown. We measured the longitudinal development of cortex from birth to six months of age using multimodal quantitative imaging of cortical microstructure. Here we show that infants' cortex undergoes profound microstructural tissue growth during the first six months of human life. Comparison of postnatal to prenatal transcriptomic gene expression data demonstrates that myelination and synaptic processes are dominant contributors to this postnatal microstructural tissue growth. Using visual cortex as a model system, we find hierarchical microstructural growth: higher-level visual areas have less mature tissue at birth than earlier visual areas but grow at faster rates. This overturns the prominent view that visual areas that are most mature at birth develop fastest. Together, in vivo, longitudinal, and quantitative measurements, which we validated with ex vivo transcriptomic data, shed light on the rate, sequence, and biological mechanisms of developing cortical systems during early infancy. Importantly, our findings propose a hypothesis that cortical myelination is a key factor in cortical development during early infancy, which has important implications for diagnosis of neurodevelopmental disorders and delays in infants.

[1] Department of Psychology, Stanford University, Stanford, CA 94305, USA. [2] Center for Cognitive and Neurobiological Imaging, Stanford, CA 94305, USA. [3] Department of Psychology, University of California Los Angeles, Los Angeles, CA 90095, USA. [4] Department of Psychology, University of Marburg, Marburg 35039, Germany. [5] Center for Mind, Brain and Behavior – CMBB, Philipps-Universität Marburg and Justus-Liebig-Universität Giessen, Marburg 35039, Germany. [6] Edmond and Lily Safra Center for Brain Sciences, Hebrew University of Jerusalem, Jerusalem 91904, Israel. [7] Neurosciences Program, Stanford University, Stanford, CA 94305, USA. [8] Wu Tsai Neurosciences Institute, Stanford University, Stanford, CA 94305, USA. [9] These authors contributed equally: Vaidehi S. Natu, Mona Rosenke. ✉email: vnatu@stanford.edu; kalanit@stanford.edu

Development of the cortical neuroarchitecture during early infancy is critical for the maturation of key sensory and cognitive functions and has lifelong consequences. During the first 6 months of life is when infants acquire crucial sensory-motor capacities[1], such as color, contrast, and spatial sensitivity[2] in the visual domain, and ability to lift the head, roll, grasp, and sit in the motor domain[3]. However, we have little knowledge of the rate, sequence, and microstructural mechanisms of the development of human sensory-motor cortices that support these basic human abilities.

Present understanding of microstructural development in the cortex is gleaned from histological investigations of a handful of sensory-motor and prefrontal regions in nonhuman primates[4–9] and humans[10,11]. Prior studies suggest that in infants, primary sensory-motor cortices are more developed than the prefrontal cortex[10–16], which is involved in complex cognitive functions. Additionally, histological research suggests that while the cortex proliferates in infancy by growing synapses[6,8,11], dendrites[5,7], axons[5,7], and myelin[10], it also prunes irrelevant connections and synapses[5,7,11,17,18]. However, there is an intense debate regarding the relative effects of microstructural growth and pruning and if they vary across cortical regions.

Critically, generalization of developmental findings from nonhuman primates to humans is tenuous as human development is longer than other species and the human visual cortex contains additional areas[19], as well as additional gyri and sulci than nonhominid primate species[20]. Moreover, different cortical areas have unique cytoarchitecture and myeloarchitecture[21,22] for example, the primary visual cortex—V1—has a unique cytoarchitecture (Stria of Gennari) already at birth[16]. Thus to achieve advancement in understanding the development of sensory systems in humans it is necessary to study the development of multiple brain areas within a human cortical system.

To fill these glaring gaps in knowledge, we leveraged advancements in quantitative magnetic resonance imaging (qMRI)[23–25] and diffusion MRI (dMRI)[26] to develop in vivo methodologies that are optimized for the infant brain. Up till now, only a handful of studies have used these methods to examine brain development[13–15,27] focusing predominantly on white matter development[14,28–30]. Quantitative measurements of proton relaxation time ($T_1$, which depends on the physiochemical tissue environment) from qMRI and mean diffusivity (MD, which depends on the density and structure of tissue through which water diffuses) from dMRI enable quantifying and longitudinally measuring the amount of brain tissue within a voxel (3D pixel in an MRI image, 1–2mm on a side) related to the neuropil[31] and myelin[32]. Thus, these quantitative metrics provide noninvasive means to inform about microstructural changes, as well as disambiguate developmental hypotheses as $T_1$ and MD are lower in tissue with denser microstructure[25,31,32]. We predicted that if cortical microstructure proliferates, $T_1$ and MD will decrease during infancy, but if microstructure is pruned, $T_1$ and MD will increase. We tested these hypotheses in (i) primary sensory-motor areas to relate our measurements to prior histological studies, and (ii) across visual areas spanning two processing hierarchies[33]. This offers an exciting opportunity to investigate the sequence and rate of microstructural development across an entire cortical system with precision and fine granularity.

Anatomical MRI, qMRI, and dMRI data were collected in 13 full-term infants (six female) who were scanned during normal sleep at 0 months [8–37 days], 3 months [78–106 days], and 6 months [167–195 days] (10 infants per timepoint, seven infants scanned longitudinally at all timepoints, Supplementary Fig. 1, Supplementary Table 1, and Methods). For quality assurance, we (i) monitored in real-time each infant's motion via an infrared camera, (ii) assessed the quality of brain images immediately after

acquisition, and (iii) repeated scans with motion artifacts. From anatomical MRIs, we generated the cortical surface for each infant and timepoint (Supplementary Fig. 1). Cortical surface reconstruction enabled achieving the most precise measurements by (i) analyzing $T_1$ and MD data in each infant's native cortical space, and (ii) using cortex-based alignment to delineate known cortical areas[34,35] in each infant's brain, which we validated maintain the same structural-functional coupling in infants (Methods).

We report three main findings. First, we show that infants' cortex undergoes profound microstructural tissue growth in primary sensory-motor cortices, during the first 6 months of human life. Second, within the visual cortex, we find hierarchical microstructural growth whereby higher-level visual areas have less mature tissue at birth than earlier visual areas but grow at faster rates. Third, comparisons of postnatal to prenatal transcriptomic gene expression data reveal that myelination and synaptic processes may be dominant contributors to this postnatal microstructural tissue growth. These findings highlight that cortical myelination is an important factor in cortical development during infancy and substantially advance our understanding of brain development during infancy at the sensory system level.

## Results

**Primary sensory cortices undergo exuberant microstructural tissue growth during the first 6 months of life.** Longitudinal cortical $T_1$ maps reveal that $T_1$ decreases from birth to 6 months of age. This decrease is heterogeneous across the cortex (example infant: Fig. 1a; all infants: Supplementary Fig. 2). For example, at 3 months, the occipital cortex and the central sulcus have lower $T_1$ (black arrows in Fig. 1a) than parietal and frontal cortices (red arrows in Fig. 1a, Supplementary Fig. 2) even as the entire cortex in 3-month-old infants has lower $T_1$ than that in newborns.

Next, we quantitatively measured $T_1$ and MD in four primary sensory-motor areas[31]: visual (V1), auditory (A1), somatosensory (S1), and motor (M1) (Fig. 1b), which overlap the cortical expanse showing rapid development. We found a systematic decrease in the distribution of $T_1$ values from birth to 6 months in all primary sensory-motor areas (Fig. 1c). Results also revealed a significant linear decrease in mean $T_1$ across all areas (linear mixed model (LMM) slopes: −1.3 to −2 [ms/day], $Ps < 10^{-7}$, Supplementary Table 2, all stats; right hemisphere: Fig. 1d, left hemisphere: Supplementary Fig. 3). Across all primary sensory-motor regions, mean $T_1$ substantially decreased from $2.03 \pm 0.07$s (mean ± standard deviation (SD)) in newborns, to $1.87 \pm 0.08$s at 3 months, to $1.74 \pm 0.06$s at 6 months. These reductions were observed in each individual infant across timepoints (Supplementary Fig. 3). Analysis of MD in these areas revealed similar significant, linear decreases in MD from 0 to 6 months (LMM slopes, rate of MD change: $-9.36 \times 10^{-7}$ to $-1.01 \times 10^{-6}$ [mm² s⁻¹/day], $Ps < 0.001$, Supplementary Table 3, all stats; right hemisphere: Fig. 1e, f, left hemisphere: Supplementary Fig. 4), which are evident in individual participants (Supplementary Fig. 4b).

**Hierarchical and heterogeneous development of cortical microstructure in visual streams.** We next used the visual cortex as a model system to investigate microstructural development as it is the best understood cortical system and contains well-defined hierarchical processing streams. The ventral visual stream[33] is involved in visual recognition, and the dorsal stream[33] is involved in visually guided actions and localization. In each infant and timepoint, we identified nine visual areas in the dorsal stream (V1d to IPS3) and eight in the ventral stream (V1v to PHC2) by using cortex-based alignment to project the Wang atlas[34] of retinotopic visual areas into each individual infant's brain at each

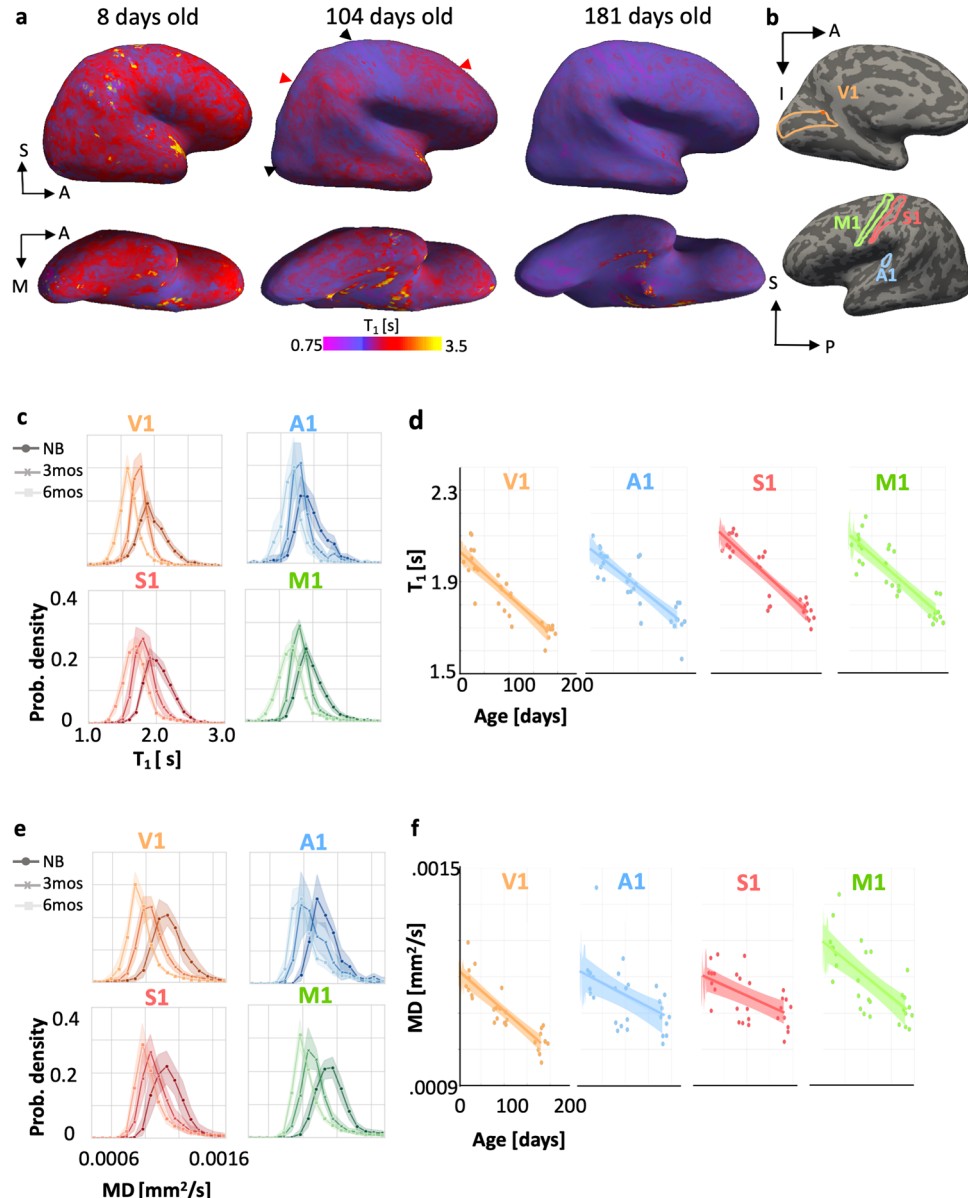

**Fig. 1 Primary sensory cortices are not fully developed at birth but show extensive microstructural tissue growth during the first 6 months of life.** **a** Right hemisphere sagittal (top) and ventral-temporal (bottom) $T_1$ maps in units of seconds [s] displayed on an inflated cortical surface of an example infant across time. *Left to right:* cortical $T_1$ at 8 days (newborn), 104 days (~3 months), and 181 days (~6 months) of age (*red/yellow:* higher $T_1$; *purple:* lower $T_1$). **b** Primary sensory-motor areas[35]: V1, A1, M1, and S1 shown on the cortical surface of this infant. **c** $T_1$ distributions across voxels of each area show a leftward shift from newborns (darker colors) to 6-month-olds (lighter colors). *Solid lines:* mean distribution; *Shaded region:* standard error of the mean across 10 infants at each timepoint. *NB:* newborn; *3 mos:* 3-month-old; *6 mos:* 6-month-old. *Darker colors* indicate younger infants. **d** $T_1$ linearly decreases with age in primary sensory-motor areas. *Each dot:* mean $T_1$ per area per infant. *Line:* Linear mixed model line fit. **e-f** Same as in c,d for mean diffusivity (MD). Shaded portions represent 95% confidence intervals. For panels **c-f**, $N_{total} = 30$, 10 infants at each time point (newborn, ~3-month-old, ~6-month-old).

timepoint. Then, we measured $T_1$ and MD in each infant's visual areas and timepoints.

Results reveal that within the first 6 months of life, $T_1$ decreases in the visual cortex on average by 0.36–0.54s in both the ventral and dorsal streams regions (Fig. 2a, b, d, e). Linear mixed modeling (LMM) per area quantified this development, revealing two main findings. First, $T_1$ significantly decreases in all dorsal ($T_1$ change/slopes: $-3$ to $-2$ [ms/day], $Ps < 10^{-7}$, Fig. 2b) and ventral visual areas (slopes: $-1.9$ to $-1.5$ [ms/day], $Ps < 10^{-6}$, Fig. 2e, Supplementary Table 4, all stats). Second, LMM estimates of $T_1$ at birth showed a systematic increase in $T_1$ at birth ascending the hierarchy of each processing stream. In the dorsal stream, $T_1$ at birth increases from V1d [2.0 ± 0.028s] to IPS3

[2.29 ± 0.027s] (Fig. 2c-right hemisphere, Supplementary Fig. 5-left hemisphere). In the ventral stream, it increases from V1v [2.01± 0.026s] to PHC2 [2.21±.027s] (Fig. 2f-right hemisphere, Supplementary Fig. 5-left hemisphere). In both streams, we observed that $T_1$ at birth plateaued before the final retinotopic area of each stream. That is, estimated $T_1$ at birth plateaued at IPS1 in the dorsal stream and at VO1 in the ventral stream.

To test if there are developmental differences across visual areas of each stream and hemisphere, we ran additional LMMs across $T_1$ data from all visual areas of a stream, with factors of age [in days], area, and hemisphere. The model comparison showed that an LMM that allows both the intercepts and slopes to vary across areas best fit the data (Methods). In both ventral and

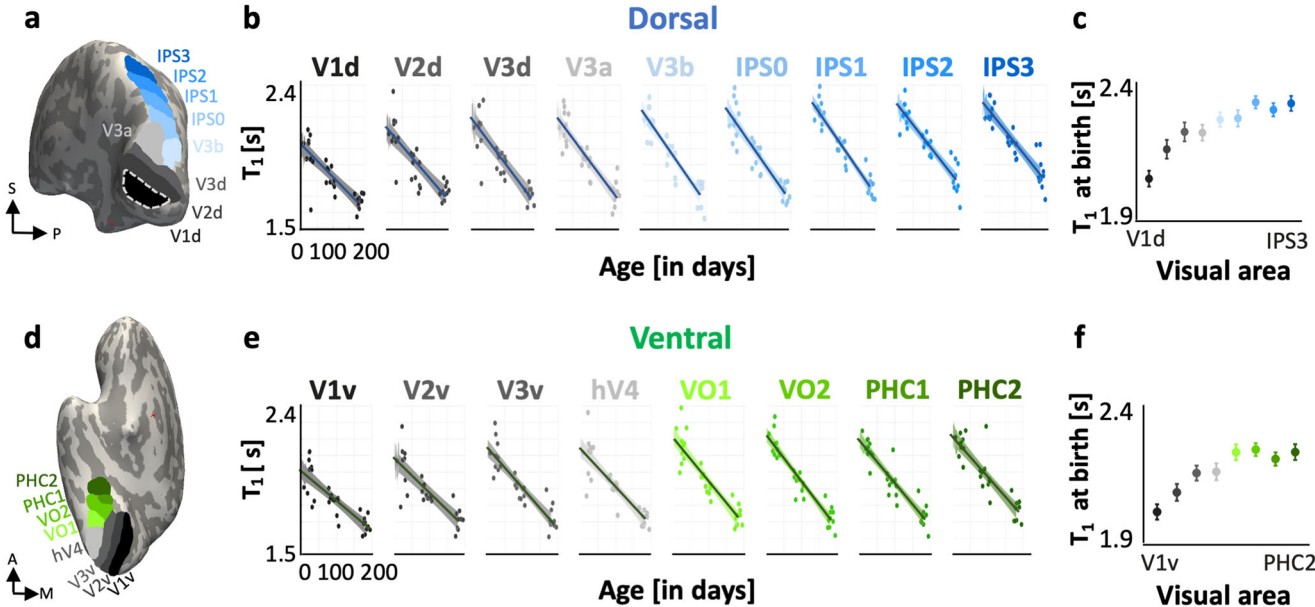

**Fig. 2 Hierarchical development of cortical microstructure in visual streams. a, d** Inflated cortical surface of an example 6-month-old infant showing nine dorsal (**a**) and eight ventral (**d**) visual areas[34]. **b, e** $T_1$ as a function of infant age in each visual area of dorsal (**b**) and ventral (**e**) visual processing streams. The color indicates the visual area (see **a, d**). *Each dot*: mean $T_1$ per area per infant. *Solid lines*: Linear mixed model (LMM) estimates of $T_1$ development for each visual area. Shaded portions represent 95% confidence intervals. **c, f** LMM estimates of mean $T_1$ at birth (LMM intercept) in each dorsal (**c**) and ventral (**f**) visual area. *Error bars*: standard error on estimates of intercepts. Data shown are of the right hemisphere. Left hemisphere data are shown in Supplementary Fig. 5. For panels **b, e**, $N_{total} = 30$, 10 infants at each time point (newborn, ~3-month-old, ~6-month-old).

dorsal streams, $T_1$ significantly varied with age ($ts > 5.13$, $Ps < 4.1 \times 10^{-7}$) and visual area ($ts > 2.75$, $Ps < 6.174 \times 10^{-3}$), but there were no differences across hemispheres ($ts < 0.722$, $Ps > 0.47$). To test if there are developmental differences between the dorsal and ventral streams, we fit another LMM across $T_1$ data of all visual areas spanning both streams with factors of age, area, and stream. $T_1$ significantly varied with age ($t_{1012} = 9.877$, $P = 4.95 \times 10^{-22}$) and area ($t_{1012} = 5.63$, $P = 2.38 \times 10^{-8}$), but there were no differences across streams ($t_{1012} = 0.89$, $P = 0.36$) and no interactions ($ts < 1.5$, $Ps > 0.12$). Together these data show that both birth values and development of $T_1$ are heterogeneous across areas of visual processing streams.

Similar results were observed with MD: (i) MD decreases from newborns to 6-month-old infants, and (ii) estimated MD at birth systematically increases from V1 to later visual areas of each stream (Supplementary Figs. 6, 7, Supplementary Table 5). LMM of MD data across areas of a stream with factors of age, area, and hemisphere showed that in both streams, MD significantly decreased with age ($ts > 3.82$, $Ps < 1.50 \times 10^{-4}$) and did not significantly vary across hemispheres ($ts < 1.37$, $Ps > 0.17$). MD development varied significantly across visual areas of the dorsal stream ($t_{496} = 3.38$, $P = 7.71 \times 10^{-4}$); there was a nonsignificant trend in the ventral stream ($t_{440} = 1.74$, $P = 0.08$). Comparing MD development across the ventral and dorsal streams, shows significant MD development with age ($t_{944} = 10.30$, $P = 1.13 \times 10^{-23}$), with differential development across areas ($t_{944} = 6.04$, $P = 2.13 \times 10^{-9}$) and streams ($t_{944} = 2.18$, $P = 0.02$) as MD decreases more in the dorsal than ventral stream in early infancy.

Combined, these data reveal intriguing properties of cortical development in the infant visual system. First, ascending visual hierarchies from V1 to higher-level visual areas, the cortical microstructure is gradually less mature at birth. Second, age-related decreases in $T_1$ and MD support the hypothesis of microstructural tissue growth in the cortex during early infancy. Nonetheless a key question remains: what microstructural tissue compartments underlie this systematic postnatal cortical tissue growth?

**Gene expression data reveals that myelination and synaptic processes are dominant mechanisms during early infancy.** To answer this question, we leveraged the transcriptomic gene expression database of postmortem tissue samples of the Brain-Span Atlas (https://www.brainspan.org) to identify candidate genes that show differential expression levels postnatally vs. prenatally. We reasoned that birth is a key developmental stage and genes that are expressed more in the cortex postnatally than prenatally may contribute to the cortical development that we observed. We examined gene expression in brain tissue samples that closely matched our in vivo data in age (postnatal samples only) and anatomical location. Thus, we used tissue samples from primary sensory-motor cortices (M1, V1, A1, V1) and higher-level visual cortices (inferior parietal cortex, superior, and inferior temporal cortex, Supplementary Tables 6, 7). For each sample, we extracted RNA-Seq expression data in Reads Per Kilobase Million (RPKM) and determined which genes show higher postnatal vs. prenatal expression along with the expression fold change (FC).

This differential analysis generated a list of several thousand genes that are expressed more in these cortical expanses postnatally than prenatally. To determine the most differentially expressed genes, we selected the genes with the largest expression fold changes (FC > 4) and assessed their significance after Bonferroni correction for multiple comparisons ($P < 5.7 \times 10^{-6}$). Figure 3a shows the expression level (per cortical sample/age) of 95 genes, which survived these criteria and Fig. 3b shows their FC in descending order. For instance, for the top 10 differentially expressed genes, expression levels increase from ~1 RPKM, prenatally, to more than six RPKM, postnatally (Fig. 3a). Intriguingly, the top-most differentially expressed gene in primary sensory-motor and visual cortices is *myelin basic protein* (*MBP*), a gene associated with myelin generation and myelin sheath wrapping[36]. It is also interesting that several other myelin-related genes, including *myelin-associated oligodendrocyte basic protein* (*MOBP*), *myelin-associated glycoprotein* (*MAG*), and *proteolipid protein 1* (*PLP-1*), are also among the top 10 most expressed genes postnatally (Fig. 3a).

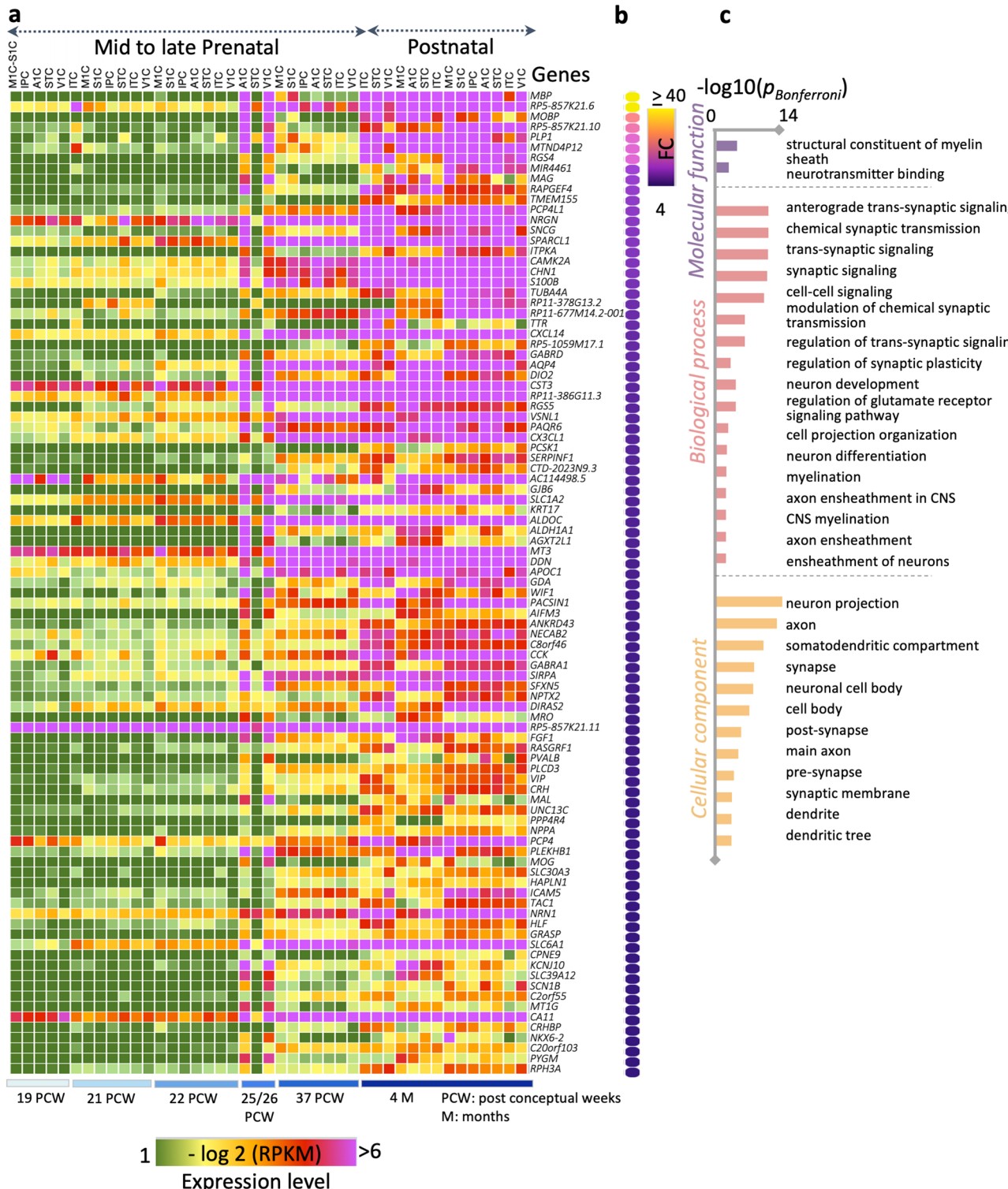

**Fig. 3 Transcriptomic gene analysis of cortical samples reveals that myelination and synaptic processes are cellular mechanisms of postnatal development. a** Matrix showing gene expression levels in prenatal (19 post conceptual weeks (pcw) to 37 pcw) and postnatal (4-months-old) cortical tissue samples for the 95 most differentially expressed genes. $N_{prenatal\ tissue\ donors} = 7$; $N_{postnatal\ tissue\ donors} = 3$. Sample demographics in Supplementary Table 6. *Rows:* genes, *columns:* cortical area (acronyms in Supplementary Table 7); *color:* expression level in reads per kilobase million (RPKM, see colorbar). **b** Gene expression fold change (FC) between postnatal vs. prenatal cortical samples of the 95 most differentially expressed genes. **c** Gene enrichment analysis showing the molecular and biological processes and cellular components related to these 95 genes.

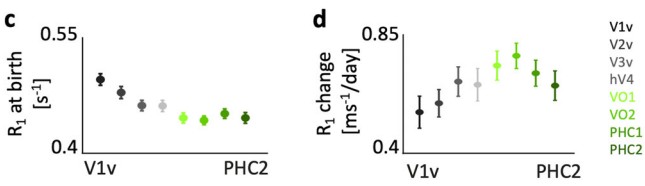

**Fig. 4 Both $R_1$ at birth and change in $R_1$ per day vary systematically across visual processing streams.** Linear mixed modeling (LMM) of $R_1$ as a function of age (Supplementary Fig. 8) provide estimates of $R_1$ at birth (LMM intercept) and rate of $R_1$ development (LMM slope) for each visual area in the dorsal (**a**, **b**) and ventral (**c**, **d**) visual streams. *Error bars:* standard error on estimates of intercepts and slopes. Data shown are from right hemisphere; Left hemisphere data are shown in Supplementary Fig. 8.

To further elucidate the molecular and cellular pathways linked to these 95 genes, we used the ToppGene toolbox (https://toppgene.cchmc.org)[37] to map this list of expressed genes to the enriched physiological processes. Comparing the 95 most significantly expressed genes to all protein-coding genes as the background set, ToppGene reported significant enrichment of several biological processes related to: (i) myelination ($P_{Bonferroni\_corrected\ (BC)} = 4.44 \times 10^{-3}$), (ii) structural constituents of myelin sheath ($P_{BC} = 2.05 \times 10^{-5}$), (iii) axonal ensheathing ($P_{BC} = 4.69 \times 10^{-3}$), (iv) synaptic signaling ($P_{BC} = 5.36 \times 10^{-12}$), and (v) cellular components of dendritic trees and spines ($P_{BC} = 3.33 \times 10^{-4}$) (Fig. 3c, Supplementary Table 8). These processes remained enriched in a control analysis in which we compared this list of top 95 most expressed genes to a different background gene set that was restricted to markers of cortical cells (neurons, astrocytes, endothelial cells, microglia, oligodendrocytes[38]).

To test if the expression of *MBP* also occurs specifically for visual areas, we conducted an additional analysis of the differential expression of genes postnatally vs. prenatally using only the samples that contained visual areas (V1, parietal and temporal expanses overlapping visual regions of the dorsal and ventral visual streams). Results showed that the top-most differentially expressed gene in the visual cortex postnatally is *MBP*. Several other myelin-related genes, including *MOBP*, *MAG*, and *PLP-1*, are also among the top 20 most expressed genes in the visual cortex postnatally. As in our main findings, ToppGene also reported significant enrichment of several biological processes related to structural constituents of myelin sheath ($P_{BC} = 1.75 \times 10^{-3}$), synaptic signaling ($P_{BC} = 1.57 \times 10^{-17}$), and cellular components of dendritic trees and spines ($P_{BC} = 1.24 \times 10^{-3}$).

**Early visual areas are more myelinated at birth, but myelinate at slower rates than later visual areas.** Transcriptomic gene analyses indicated that myelination, synaptogenesis, and dendritic processes as key microstructural mechanisms developing after birth. As myelin is known to decrease both $T_1$ relaxation time[24,25,31,32,39] and MD[26], these data suggest that cortical myelination may contribute to the reduction in $T_1$ and MD from birth to 6 months observed in the cortex. In turn, this raises an

intriguing possibility that these metrics might inform about the sequence and rate of cortical myelination in infancy. While MD depends on myelin, it also depends on axons' radii and packing. As such, MD varies in complex nonlinear ways with myelin. Hence, it is difficult to make inferences from MD about cortical myelin[40]. $T_1$ is inversely related to myelin fraction[39] ($T_1 \sim \frac{1}{myelin\ fraction}$). Because of the inverse relationship, a change in $T_1$ depends both on the change in myelin and the initial myelin fraction in the voxel[41]. Consequently, a similar change in myelin would produce a larger change in $T_1$ in voxels that are less myelinated (smaller myelin fraction) than those that are more myelinated (larger myelin fraction). In contrast to $T_1$, tissue relaxation rate $R_1$ ($R_1 = \frac{1}{T_1}$) varies linearly with myelin[39]. Therefore, changes in $R_1$ are linearly related to myelin changes, independent of a voxel's myelin fraction. Hence, to glean insights about cortical myelination in early infancy, we measured $R_1$ in each infant, timepoint, and visual area.

Results revealed linear increases in $R_1$ in both the ventral and dorsal visual processing streams and in both hemispheres (Supplementary Fig. 8). We used LMMs to evaluate $R_1$ at birth and change in $R_1$ per day across the first 6 months of life. We found differences across visual areas of $R_1$ at birth (Fig. 4a, c). Using a random intercept/random slope LMM relating the mean $R_1$ to participants' age [in days] across all areas of a stream (Methods), we found a significant effect of area across both streams and hemispheres ($ts > 5.00$, $Ps < 1.102 \times 10^{-8}$). Like $T_1$ at birth (Fig. 2), which varied progressively across the visual hierarchy, $R_1$ at birth is highest in V1 ($R_1 = 0.496 \pm 0.008$) and progressively decreased from early to later visual areas across both the visual streams (Fig. 4a, c). Indeed, $R_1$ at birth is numerically lowest in right IPS1 ($R_1 = 0.428 \pm 0.005$) and left IPS2 ($R_1 = 0.429 \pm 0.005$) in the dorsal stream, and left and right VO2 ($R_1 = 0.445 \pm 0.005$) in the ventral stream (Supplementary Table 9).

Changes in $R_1$ during the first 6 months become progressively larger from V1 (right V1d: $0.59 \pm 0.06$ [ms$^{-1}$/day]) to IPS0 in the dorsal stream (right IPS0: $0.76 \pm 0.04$ [ms$^{-1}$/day], Fig. 4b) and from V1 (right V1v: $0.56 \pm 0.06$ [ms$^{-1}$/day]) to VO2 in the ventral stream (right VO2: $0.76 \pm 0.04$ [ms$^{-1}$/day], Fig. 4d, Supplementary Table 9, for all stats). While $R_1$ in IPS0 and VO2 develop ~28% faster than in V1 during the first 6 months of life, the highest retinotopic areas in both streams showed a lower rate of $R_1$ change. That is, the rate of change in $R_1$ in IPS2/3 is similar to that of V2d (Fig. 4b) and that of PHC1/2 is similar to V3v (Fig. 4d).

To quantify the development in $R_1$ across areas of each visual processing stream, we fit LMMs relating $R_1$ across all visual areas of a stream to participants' age [in days], area, and hemisphere. The model comparison showed that an LMM which allows both intercepts and slopes to vary across areas best fit the data. Results reveal that in both the dorsal and ventral streams, $R_1$ increased with age ($ts > 6.01$, $Ps < 3.67 \times 10^{-9}$) and its development significantly varies across areas ($ts > 2.60$, $Ps < 9.53 \times 10^{-3}$), but not hemispheres ($ts < 0.877$, $Ps > 0.38$, Supplementary Table 9). To test differences between $R_1$ development across the two streams, we ran another LMM on all visual areas with factors of age, area, and stream. Results reveal that $R_1$ significantly varied with age ($t_{1012} = 11.21$, $P = 1.30 \times 10^{-27}$) and area ($t_{1012} = 4.81$, $P = 1.65 \times 10^{-6}$), but there was no difference in $R_1$ development across streams ($t_{1012} = 0.82$, $P = 0.40$) or interactions ($ts < 1.64$, $Ps > 0.1$). These results suggest that $R_1$ at birth and $R_1$ development vary across visual areas of ventral and dorsal visual processing streams.

**Discussion**
Combining in vivo metrics that have meaningful units with gene expression analyses, we show that during the first 6 months of

life, infants' cortex undergoes exuberant microstructural tissue growth related to myelination, synaptogenesis, and dendritic processes. Within the visual cortex, we find hierarchical development across two processing streams, where earlier regions are more mature at birth, but develop slower than later visual areas. As $R_1$ is linearly related to myelin, and cortical iron is relatively negligible in early infancy[42], our data suggest that in both processing streams, earlier visual areas have higher myelin content at birth, but they myelinate at slower rates than later retinotopic areas up to IPS0/1 in the dorsal stream and VO2 in the ventral stream.

Our data show strong effects of cortical increases in $R_1$ and decreases in MD supporting microstructural tissue growth in sensory-motor cortices from 0 to 6 months of age. These findings: (i) are supported by our transcriptomic analyses revealing myelination, synaptogenesis, and dendritic processes as key mechanisms of infants' sensory-motor cortex development, and (ii) are consistent with prior histological data showing synaptogenesis[11] and dendritic growth[7] in primate V1 and A1 during early infancy. While we do not find evidence for pruning, due to the complexity of factors that impact $R_1$ ($T_1$) in the cortex, we acknowledge that we cannot conclude from qMRI metrics alone that there is no cortical pruning during early infancy. For example, it is possible that pruning effects may not be visible to qMRI metrics if the pruned neurites are unmyelinated. Additionally, changes in iron due to phagocytosis associated with pruning[43], may modulate $R_1$ ($T_1$) but may be obscured by larger effects of cortical myelination. Finally, our results do not preclude the possibility that pruning may occur later in infant development[9,11] following this exuberant microstructural tissue growth during the first 6 months of life. However, our transcriptomic data provide complementary evidence for the growth of multiple cortical tissue components during early infancy. Future histological investigations in pediatric samples containing sensory-motor cortices can elucidate precisely which cellular components develop during infancy in each visual area.

Gene analyses reveal that *MBP* is the top-most differentially expressed gene after birth, and both ex vivo and in vivo studies show that cortical $R_1$ increases with higher myelin content[25,32]. Consequently, this suggests that myelination contributes to tissue changes in infants' cortex measured with $R_1$. These findings are transformative for three reasons:

First, they uncover the development of cellular and biological mechanisms in infant sensory-motor cortices. While research in other species also revealed postnatal myelin growth and synaptogenesis[9,10], as well as initial overproduction of axons[5,44], generalizing across species is challenging as species vary in brain size, developmental trajectory[45], and cortical functional organization. Hence, to uncover the development of the human brain, it is critical to study the trajectory of biological mechanisms in humans, as well as between brain regions across the human lifespan[46].

Second, our findings link specific biological mechanisms to in vivo MRI measurements. Moreover, microstructural development likely precedes or occurs together with functional development, giving us an opportunity to track the earliest developments in the cortex. This is important as it lays a necessary foundation for developing in vivo markers for diagnosing typical and atypical brain development. As babies in the first year of life are a highly vulnerable population, the ability to use in vivo tools to diagnose neurodevelopmental disorders and deficiencies, as well as to intervene early could not be of greater importance.

Third, our findings suggest that not only synaptogenesis[6,11,47] and neurite sprouting[7] but also cortical myelination is critical for the development of brain function and ultimately behavior. As cortical myelination is thought to be activity dependent[48–50],

these data further suggest that increases in cortical $R_1$ during early infancy may be linked to functional changes in the same cortical areas. This opens exciting avenues for future research in infants that will examine the link between cortical myelination and functional brain development and suggests that cortical myelination should be considered in any future infant research[51–53].

Additionally our study advances understanding of the development of the human visual system during early infancy by investigating the development of multiple visual areas across two main processing streams. This expands on the prior research that used V1 as a model system to study development[6,11] despite its unique cytoarchitecture[54–56] and myeloarchitecture[16].

Our data across the visual system reveal three important findings. First, contrary to the theory that V1 is myelinated at birth[16,57], our analyses of $R_1$ suggest that while V1 is more developed than other visual areas at birth, it continues to develop and profoundly myelinates during the first 6 months of life. An interesting question for future research is whether the earlier maturation of V1 is due to overall higher myelination across cortical layers or is specific to the earlier myelination of the Stria of Gennari, which is highly myelinated in adults[58]. Laminar analyses in histological pediatric samples will be instrumental for answering this open question. Second, contrary to the hypothesis that primary sensory-motor areas (such as V1) also myelinate the fastest postnatally[16,57], analysis of the rate of $R_1$ development during the first six months of life reveals that in the human visual system V1 does not myelinate the fastest. In fact, $R_1$ in several visual areas catches up with the $R_1$ of V1 by 6 months of age, even though they have lower values at birth. Third, we find that microstructural development varies across the visual hierarchy. That is, at birth higher visual areas of both ventral and dorsal streams have lower $R_1$, and thus, are likely less myelinated than early retinotopic areas. In contrast, the rate of $R_1$ (and consequently myelin) development is progressively faster in later than early visual areas. This progression is hierarchical only up to a certain point of each processing stream—VO1/2 in the ventral stream (Fig. 4b) and IPS0/1 in the dorsal stream (Fig. 4d) after which it decreases. Consequently, by 6 months of age, $R_1$ in several retinotopic visual areas catches up with the $R_1$ of V1 (Supplementary Fig. 8). This trajectory of microstructural growth across visual processing hierarchies may reflect an interplay between sensory experience and the development of cortical systems. We hypothesize that the more mature primary sensory-motor cortices at birth may provide scaffolding for the development of cortical systems, but the sensory richness of the postnatal environment may accelerate activity-dependent myelination and synaptogenesis in higher-level sensory cortices[6,48,49,59].

These findings also open questions for future research. One question is whether the microstructural changes observed here are linked to changes in visual function during infancy. Recent research suggests that there is some level of retinotopic functional connectivity in early infancy[60] and that by 5 months of age there is some level of retinotopic organization in early and intermediate human visual areas[61] (V1, V2, V3, V3ab, V4). As neural receptive fields that underlie these maps develop during infancy[62,63] and continue to develop during childhood[64,65], our data suggest the possibility that microstructural development of retinotopic visual areas found here will be coupled with fine-tuning of receptive fields and retinotopic maps in the infant brain. Future research may determine if such neural fine-tuning is also associated with the rapid development of visual-spatial acuity[66] and contrast sensitivity[67] during early infancy.

In conclusion, our findings necessitate a rethinking of how cortical microstructure in sensory systems develops in infants, and open avenues to examine the impact of cortical myelination on the development of brain function. Young infants are a highly

vulnerable population for which the ability to diagnose delayed and atypical development could not be of greater importance. These powerful multimodal methodologies enable noninvasive, longitudinal measurements within individual infants that are linked to specific biological mechanisms. Thus, our study has important implications for identifying neurodevelopmental delays and disorders in infants which may lead to early inventions and better life-long outcomes.

## Methods

**Participants.** Sixteen full-term and healthy infants (seven female) were recruited to participate in this study. Three infants provided no usable data because they could not stay asleep once the MRI sequences started. Here, we report data from 13 infants (6 female) across three age timepoints: newborn/0 months [8–37 days], 3 months [78–106 days], and 6 months [167–195 days], with 10 participants per timepoint. Two participants were reinvited to complete scans for their 6-months session that could not be completed during the first try. Both rescans were performed within 7 days and participants were still within the age range for the 6-month timepoint. The participant population was racially and ethnically diverse reflecting the population of the San Francisco Bay Area, including two Hispanic, nine Caucasian, two Asian, three multiracial (two Asian and Caucasian; one Native Hawaiian or Other Pacific Islander) participants. Seven out of these 13 infants participated in MRI in all three timepoints (0, 3, and 6 months). Due to the Covid-19 pandemic and restricted research guidelines, data acquisition was halted. Consequently, the remaining infants participated in either 1 or 2 sessions. Participation of the 13 infants whose data are reported in this study is summarized in Supplementary Table 1.

*Expectant mother and infant screening procedure.* Expectant mothers and their infants in our study were recruited from the San Francisco Bay Area using social media platforms. We performed a two-step screening process for expectant mothers. First, mothers were screened over the phone for eligibility based on exclusionary criteria designed to recruit a sample of typically developing infants, and second, eligible expectant mothers were screened once again after giving birth. Exclusionary criteria for expectant mothers were as follows: recreational drug use during pregnancy, alcohol use during pregnancy (more than three instances of alcohol consumption per trimester; more than one drink per occasion), lifetime diagnosis of autism spectrum disorder or a disorder involving psychosis or mania, taking prescription medications for any of these disorders during pregnancy, insufficient written or spoken English ability to comprehend study instructions, and learning disabilities. Exclusionary criteria for infants were preterm birth (<37 gestational weeks), low birth weight (<2.49 kgs), small height (<45 cms), any congenital, genetic, or neurological disorders, visual problems, complications during birth that involved the infant (e.g., NICU stay), history of head trauma, and contraindications for MRI (e.g., metal implants).

**MRI (magnetic resonance imaging) Procedure.** We acquired T2-weighted MRI, quantitative MRI (qMRI), and diffusion MRI (qMRI) data for each infant and time point. Study protocols for these scans were approved by the Stanford University Internal Review Board on Human Subjects Research. Scanning sessions were scheduled in the evenings close in time to the infants' typical bedtime. The duration of each session lasted between 2.5 and 5 hours including time to prepare the infant and waiting time for them to fall asleep. Upon arrival, caregivers provided written, informed consent for themselves and their infant to participate in the study. Before entering the MRI suite: (i) both caregiver and infant were checked to ensure that they were safe and metal-free to enter the MRI suite and (ii) caregivers changed the infants into MR-safe cotton onesies and footed pants provided by the researchers. The infant was swaddled with a blanket with their hands to their sides to avoid their hands creating a loop during MRI. During scans of newborn infants, an MR-safe plastic immobilizer (MedVac, www.supertechx-ray.com) was used to stabilize the infant and their head position. Once the infant was ready for scanning, the caregiver and infant entered the MR suite. The caregiver was instructed to follow their child's typical sleep routine. As the infant fell asleep, researchers inserted soft wax earplugs into the infant's ears. Once the infant was asleep, the caregiver gently placed the infant with their head inside the head coil, and their body on a makeshift cradle on the scanner bed. This cradle was created by placing weighted bags at the edges of the bed to prevent any side-to-side movement. Finally, to lower sound transmission, MRI compatible neonatal Noise Attenuators (https://newborncare.natus.com/products-services/newborn-care-products/nursery-essentials/minimuffs-neonatal-noise-attenuators) were placed on the infant's ears and additional pads were also placed around the infant's head to stabilize head motion. An experimenter stayed inside the MR suite with the infant during the entire scan to monitor the infant and soothe them in case they woke up during scanning. For additional monitoring of the infant's safety and motion quality an infrared camera was affixed to the head coil and positioned for viewing the infant's face in the scanner. The researcher operating the scanner monitored the infant via the camera feed, which allowed for the scan to be stopped immediately if the infant showed signs of waking or distress. This setup allowed tracking

the infant's motion; scans were stopped and repeated if there was excessive head motion.

*Data quality assurance during MRI.* To ensure high data quality, in addition to real-time monitoring of the infant's motion via an infrared camera, acquired scans were assessed immediately after acquisition of each sequence and repeated if necessary. Factors for repetition included head motion detected on the infrared camera, or head motion detected on the acquired MR images reflected in blurring of otherwise detailed anatomical images and partial voluming effects. On average, 50% of all scans across infants were successful on the first try while the rest had to be repeated.

**Data acquisition.** All participants participated in multiple scans in each session to obtain anatomical MRI, qMRI, and dMRI data. Data were acquired at two identical 3T GE Discovery MR750 Scanners (GE Healthcare) and Nova 32-ch head coils (Nova Medical) located at Stanford University: (i) Center for Cognitive and Neurobiological Imaging (CNI) and (ii) Lucas Imaging Center using identical acquisition sequences and protocols. Prior work has shown that quantitative imaging metrics are replicable across scanners and subjects and have low intersite discrepancies[68,69]. As infants have low weight, all imaging was done with Normal SAR level to ensure their safety.

*Anatomical MRI. T2-weighted:* T2-weighted images were acquired for participants in each of 0, 3, 6 months timepoints. T2-weighed image acquisition parameters: TE = 124 ms; TR = 3650 ms; echo train length = 120; voxel size = $(0.8 \text{ mm})^3$; FOV = 20.5 cm; Scan time: 4 min and 5 s.

*Quantitative MRI. Spoiled-gradient echo images (SPGR):* were acquired for all participants in each of 0, 3, 6 months. These images were used together with the IR-EPI sequence to generate whole-brain synthetic T1-weighted images. We acquired 4 SPGRs whole-brain images with different flip angles: α = 4°, 10°, 15°, 20°; TE = 3 ms; TR = 14 ms; voxel size = 1 mm³; number of slices = 120; FOV = 22.4 cm; Scan time: (4:55 min) × 4.

*Inversion-recovery EPI (IR-EPI).* IR-EPI images were acquired from all participants in each of 0, 3, 6 months timepoints. We acquired multiple inversion times (TI) in the IR-EPI using a slice-shuffling technique[70]: 20 TIs with the first TI = 50 ms and TI interval = 150 ms; we also acquired a second IR-EPI with reverse phase-encoding direction. Other acquisition parameters are voxel size = $(2 \text{ mm})^3$; number of slices = 60; FOV = 20 cm; in-plane/through-plane acceleration = 1/3; Scan time:1:45 min) × 2.

*Diffusion MRI.* We obtained dMRI data from nine newborn participants, and 10 participants at 3 months and 6 months of age. One newborn woke up prior to run completion and we could not complete dMRI acquisition. dMRI parameters: multi-shell, diffusion directions/*b* value = 9/0, 30/700, 64/2000; TE = 75.7 ms; TR = 2800 ms; voxel size = $(2 \text{ mm})^3$; number of slices = 60; FOV = 20 cm; in-plane/through-plane acceleration = 1/3; scan time: 5:08 min We also acquired a short dMRI scan with reverse phase-encoding direction and only 6 *b* = 0 images (scan time 0:20 min).

**Statistics and Reproducibility.** The MRI data analysis pipeline is summarized in Supplementary Fig. 1. In brief, IR-EPI data were used to estimate $T_1$ relaxation time at each voxel. These data were also used together with the SPGRs to generate synthetic T1-weighted whole-brain anatomies of each infant at each timepoint. All data from that timepoint were aligned to this anatomical image. T2-weighted images were used for segmentation of gray-white matter to generate cortical surface reconstructions and dMRI data and diffusional kurtosis imaging was used to estimate MD in each voxel. All infant data were kept in native space as all analyses were performed within-subject and within-timepoint.

*Quantitative $T_1$ relaxation time modeling.* The signal equation of $T_1$ relaxation of an inversion-recovery sequence is an exponential decay:

$$S(t) = a(1 - be^{-t/T1}) \qquad (1)$$

where $t$ is the inversion time, $a$ is proportional to the initial magnetization of the voxel, $b$ is the effective inversion coefficient of the voxel (for perfect inversion $b = 2$). We applied an absolute value operation on both sides of the equation and used the resulting equation as the fitting model. We use the absolute value of the signal equation because we use the magnitude images to fit the model. The magnitude images only keep the information about the strength of the signal but not the phase or the sign of the signal.

First, as part of the preprocessing, we performed susceptibility-induced distortion correction on the IR-EPI images using FSL's[71] top-up correction and the IR-EPI acquisition with reverse phase-encoding direction. We then used the distortion corrected images to fit the above $T_1$ relaxation signal model using a multi-dimensional Levenberg-Marquardt algorithm[72]. The output of the algorithm

is the estimated $T_1$ in each voxel as well as the model goodness of fit ($R^2$) value in each voxel.

*Generation of T1-weighted whole-brain anatomies.* From the SPGRs and IR-EPI scans, synthetic T1-weighted whole-brain images were generated using mrQ software (https://github.com/mezera/mrQ). We analyzed all data in the native infant space and did not align our data to any template brain. All data (T2-weighted-anatomy, quantitative $T_1$ relaxation estimates, and MRI data) from a given timepoint were aligned to this brain volume (Supplementary Fig. 1).

*Generation of cortical surfaces.* To generate cortical surface reconstructions, we used both T2-weighted and synthetic T1-weighted anatomies. We used multiple steps to generate accurate cortical surface reconstructions of each infant's brain at each timepoint. (1) An initial segmentation of gray and white matter was generated from the synthetic T1-weighted brain volume using infant FreeSurfer's automatic segmentation code developed for infant data (infant-recon-all; https://surfer.nmr.mgh.harvard.edu/fswiki/infantFS)[73]. This initial segmentation generates pial and white matter surfaces, and surfaces of curvature, thickness, and surface area. However, this initial FreeSurfer's segmentation misses portions of the infant's gray matter, as the contrast of infants' T1-weighted images were not differentiated enough between gray and white matter to generate an accurate segmentation. (2) We used T2-weighted anatomical images, which have a better contrast between gray and white matter in infants, and an independent brain extraction toolbox (Brain Extraction and Analysis Toolbox, iBEAT, v-2.0 cloud processing, https://ibeat.wildapricot.org/)[74–77] to generate more accurate white and gray matter segmentations. (3) The iBEAT segmentation was further manually corrected to fix segmentation any additional errors (such as holes and handles) using ITK-SNAP (http://www.itksnap.org/) in white matter as well as gray matter. (4) The manually corrected iBEAT segmentation was aligned to the T1-weighted anatomy that was used for the FreeSurfer segmentation using manual rigid-body alignment in ITK-SNAP (Supplementary Fig. 1). (5) The aligned and segmented volume was then reinstalled into FreeSurfer using software we developed based on infant FreeSurfer functions (https://github.com/VPNL/babies_graymatter). This process updates the white matter segmentation and the cortical surfaces in the subject's FreeSurfer directory to render the accurate surfaces (Supplementary Fig. 1). This accurate surface was used for visualization, and cortex-based alignment with cortical atlases[34,35,78,79].

*DMRI.* DMRI data were preprocessed using a combination of tools from mrTrix3 (https://github.com/MRtrix3/mrtrix3)[80] and mrDiffusion toolbox (http://github.com/vistalab/vistasoft). (1) We denoised the data using a principal component analysis[81]. (2) We used FSL's top-up tool (https://fsl.fmrib.ox.ac.uk/) and one image collected in the opposite phase-encoding direction to correct for susceptibility-induced distortions. (3) We used FSL's eddy to perform eddy current and motion corrections. Motion correction included outlier slice detection and replacement[82]. (4) We performed bias correction using ANTs[83]. (5) These pre-processed dMRI images were registered to the whole-brain T2-weighted anatomy using whole-brain rigid-body registration in a two-stage model with a coarse-to-fine approach that maximized mutual information. (6) mrTrix3 software was used to fit tensors to each voxel using a least-squares algorithm that removes outliers. From the diffusional kurtosis tensor files, we estimated mean diffusivity (MD) maps in each voxel of the brain.

*dMRI quality assurance.* Out of the 29 dMRI acquisitions, one newborn acquisition was missing the reverse phase-encoding image required for susceptibility correction. Thus, we report data from eight newborns, ten 3-month-old, and ten 6-month-old infants. Across all acquisitions, less than $5.0 \pm 0.7\%$ of dMRI images were identified as outliers by FSL's eddy tool. We found no effect of age across the outliers (no main effect of age: $F_{2,25} = 2.84$, $P = 0.08$, newborn: $1.20 \pm 0.83$; 3 months: $0.4 \pm 0.40$; 6 months: $0.67 \pm 0.85$) suggesting that the developmental data were well controlled across all ages of infants.

**Delineation of primary sensory cortices and ventral and dorsal visual areas.** Our goal was to examine developmental changes in quantitative $T_1$ and MD in the gray matter of four primary sensory-motor cortices as well as across the ventral and dorsal visual processing streams from birth to 6 months of age. To delineate these regions in infants, we used brain atlases available on the FreeSurfer adult average brain which were projected to each individual participant's cortical surface at each timepoint using infant FreeSurfer's cortex-based alignment algorithm. Without functional data, cortex-based alignment is the most accurate method for defining brain areas in individual brains from atlases[84]. Critically, the major sulci and gyri that are used for cortex-based alignment are present at birth[85] and are visible on the cortical surface reconstructions of each of our infants. We used the Glasser atlas[35] to delineate the primary visual (V1), primary auditory (A1), primary motor (M1), and primary somatosensory (S1) cortices. We used the Wang atlas[34] to delineate 9 regions spanning the dorsal visual stream (V1d, V2d, V3d, V3a, V3b, IPS0, IPS1, IPS2, and IPS3) and 8 regions spanning the ventral visual stream (V1v, V2v, V3v, hV4, VO1, VO2, PHC1, and PHC2) in each participant's brain.

*Quality check on the delineation of regions of interest (ROIs).* Typically for functional and anatomical analysis in adults, brain atlases generated using average adult brains are used for delineating regions of interest using cortex-based alignment to the cortical surface of each individual subject. For instance, when using the Wang atlas[34] to delineate the primary visual area, V1, which lies in the calcarine sulcus, we expect that cortical-based alignment of V1 from the Wang atlas to the individual brain should map V1 accurately to the calcarine sulcus of the individual cortical surface. As we used cortex-based alignment and atlases developed for the adult brain to define areas in the infant brain, we first sought to test how well anatomical ROIs from adult atlases were mapped to infants' brains. To test this correspondence, we compared how the calcarine sulcus manually defined on each individual brain compares to the cortex-based aligned calcarine sulcus from the Desikan atlas[78], an anatomical parcellation, on FreeSurfer's average adult cortical surface. The manually defined calcarine sulcus was defined from the posterior end of the occipital pole to the prostriate (ProS)—a region along the anterior bank of the parietal-occipital sulcus. We chose the calcarine sulcus as the benchmark because V1 is located in the calcarine sulcus, and it can be identified anatomically in each individual brain. In each infant and timepoint ($N = 30$) and 10 adults (ages 22–27; from our previous study[32]), we calculated the overlap between the Desikan-calcarine and individual-subject-calcarine using the dice coefficient[86]. We found that the dice coefficient was $0.66 \pm 0.10$ (Mean $\pm$ SD) in infants and $0.67 \pm .05$ in adults. There were no age-related differences in dice coefficients between infants and adults (no main effect of age: $F_{1,76} = 0.02$, $P = 0.89$, 2-way analysis of variance (ANOVA) with factors of age group (infant/adult) and hemisphere (left/right) and no differences between hemispheres ($F_{1,76} = 0.02$, $P = 0.88$). This analysis suggests that cortex-based alignment of brain atlases based on adult templates to infants' cortical surfaces is similar in quality to this transformation in adults.

As V1 is relatively more developed than other areas of the cortex in newborns, we also examined the quality of cortex-based alignment in infants of two later-developing regions, parahippocampal areas PHC-1 and PHC-2. In adults, these regions are (i) located anterior to VO2 and extend across the collateral sulcus (CoS) into the posterior parahippocampal cortex (PHC)[87], and (ii) overlap with the functionally defined parahippocampal place area[87] (also referred to as CoS-places[88]). We tested if cortex-based alignment of these high-level ROIs to the infant brain will show the same structural-anatomical coupling in infants as in adults. To do so, we used cortex-based alignment to project from the FreeSurfer average cortical surface to each infant's cortical surface at each timepoint: (i) the CoS from the Desikan atlas[78], (ii) PHC-1 and 2 from the Wang atlas[34], and (iii) CoS-places from the Rosenke atlas[79]. Results shown in Supplementary Fig. 9 demonstrate similar functional-structural coupling of PHC-1 and 2 to the CoS as well as similar coupling between the PHC-1 and 2 to the CoS-places in infants. Out of the 60 hemispheres, there was a single hemisphere where the anatomical CoS was shifted more laterally and was less aligned to the infant's CoS. However, even in this hemisphere, PHC-1 and 2 and CoS-Places were still aligned to each other and the CoS. This analysis validates our delineation of ROIs in infants using cortex-based alignment of data from adult atlases.

**Analysis of mean $T_1$, MD, and $R_1$ in cortical areas and their development.** After delineating brain areas in each participant, we calculated the distribution and mean $T_1/R_1$ and mean diffusivity (MD) in each participant and timepoint. We used linear mixed models (LMMs) to determine if there were age-related changes of $T_1$ and MD within and across areas. While we expect that developmental trends will eventually asymptote over the lifespan[89,90], during the first 6 months of life, our data showed linear trends (Figs. 1 and 2) and, in general, LMM produced good fits to the data ($R^2$ in Supplementary Tables 2–5 and 9). LMMs were fit using the MATLAB 2017b function *fitlme*. To quantify the development of $T_1$, $R_1$, and MD, for each area, we fit a LMM relating the mean metric of that area to participants' age [in days]. We ran two types of LMM per area and data type: (1) LMM with a random intercept/fixed slope, which allows only the intercepts to vary across participants and accounts for the fact that the same infants participated across multiple timepoints, and (2) LMM with a random intercept/random slope, which allows both intercepts and slopes to vary across participants. Model comparison using likelihood testing revealed that the random intercept/fixed slope model fit the data best in all cases (in all model comparisons: Ps < 0.05). Thus, we report the parameters of the LMMs with random intercepts in all our analyses below:

$$mean(T_1/R_1/MD) \sim age\ of\ infant[days] + (1|infant) \qquad (2)$$

where $mean(T_1/R_1/MD)$ is the dependent variable, *age of infant* is a continuous predictor (fixed effect), and the term: *1|infant* indicates that random intercepts are used for each participant. Per model, we obtained: (1) the intercept, which estimates the values of the dependent variable of interest at birth, and (2) the average slope, which estimates the rate of development per day of that variable. The linear fits of the LMMs are plotted in Figs. 1 and 2 and Supplementary Figs. 3–6 and we report slopes and significant levels (Ps) of all areas in both hemispheres in Supplementary Tables 2–5. Since we ran LMMs for each area individually, we performed Bonferroni correction for multiple comparisons for each analysis: (1) across primary sensory-motor cortices (four areas, six comparisons), (2) across the dorsal visual stream (nine areas, 36 comparisons), and (3) across the ventral visual stream (eight areas, 28 comparisons).

We used a second set of the linear mixed models to quantify the development of $T_1$, $R_1$ and MD per stream, with factors: age, area, and hemisphere. We fit a LMM relating the mean metric ($T_1$ or $R_1$ or MD) of each area to participants' age [in days]. We ran two types of LMM per stream and metric: (1) LMM with a random intercept/fixed slope, which allows only the intercepts to vary across areas, and (2) LMM with a random intercept/random slope, which allows both intercepts and slopes to vary across areas and hemispheres. Likelihood tests comparing these two models revealed that the random intercept/random slope model fit the data best for $T_1$ and $R_1$ (in all model comparisons the latter model was significantly better than the former, likelihood test, Ps < 0.01; degrees of freedom (ventral): 472; degrees of freedom (dorsal): 532). Thus, we report the parameters of LMMs with random intercepts and random slopes for these metrics. For MD, results revealed that the random intercept and random slope model fit the data best in the dorsal stream ($P = 0.001$, degrees of freedom (dorsal): 496) but the random intercept/fixed slope model fit the data best in the ventral stream (in model comparisons the latter model was not significantly better than the former ($P = 0.87$, degrees of freedom (ventral): 440). However, for statistical consistency, for the ventral stream, we also report data from the random intercept/random slope model.

Finally, we used a third set of LMMs to test if there are any developmental differences in $T_1$, $R_1$ and MD between the dorsal and ventral streams, with factors: age, area, and stream. Here, the random intercept/random slope model best fit the data (model comparison, Ps < 0.01).

**Transcriptomic gene data analysis of postnatal versus prenatal tissue samples**. To assess what microstructural tissue compartments may be linked to the observed postnatal tissue growth in cortex related to decrease in $T_1$ and MD, we used the transcriptomic gene expression database of postmortem tissue samples made available by the BrainSpan Atlas portal (https://www.brainspan.org). We examined: (1) if there were differences in the expression levels for genes in the postnatal tissue samples as compared to the prenatal tissue samples and (2) if so, what cellular and biological processes are related to these genes of interest. In order to closely match our in vivo data, we selected the postmortem postnatal tissue samples within our in vivo age range (0 to 6 months). Prenatal samples were between 19 post conceptual weeks (pcw) to 37 pcw, which is just prior to birth. Supplementary Table 6 includes demographic details of the postnatal and prenatal samples. Within these postmortem samples, we compared to tissue from primary sensory cortices (V1, M1, S1, and A1) and parietal and temporal regions overlapping visual regions of the dorsal and ventral visual streams, respectively, to match our in vivo data (Supplementary Table 7 for the complete list of brain regions from individual prenatal and postnatal samples).

The differential analysis provides information about which genes are differentially expressed when we compared our target (postnatal) versus control (prenatal) sample sets. Specifically, differential expression-level analysis reveals a list of several thousand genes along with the gene level RNA-Sequencing expression data in reads per kilobase million (RPKM, data were log2-scaled). The analysis also estimated how many more times the genes are expressed postnatally vs. prenatally (fold changes, FC) and provides the statistical significance of the contrast (p-values). Fold change is measured as the average log2(intensity/expression) values of all samples in the target sample minus the average log2(intensity/expression) of the control samples. As standard practice[91] we applied two thresholds: (1) a threshold of fold change: FC > 4) and (2) a Bonferroni correction ($P < 5.7 \times 10^{-6}$) related to the differential analysis. Ninety-five genes passed these thresholds.

*Functional enrichment analysis*. Next, to assess what molecular and biological processes are linked to our top gene list we inputted the list of genes to a toolbox created for gene list enrichment analysis (ToppGene https://toppgene.cchmc.org)[37] Specifically, this toolbox identifies the biological pathways that are enriched (over-represented) by the expression of the genes of interest more than that would be expected by chance. Each gene is compared with the genes related to a specific pathway and a p-value of the enrichment of a pathway is computed and multiple-test correction is applied. A table of biological, molecular, and cellular processes related to this list of genes is derived from this analysis. A Bonferroni correction was applied as a multiple comparisons adjustment. Information on the biological pathways related to the top-most genes is ranked by statistical significance of functional enrichment ($-\log_{10}$ ($p$ value Bonferroni corrected). We performed this analysis first using (i) all protein-coding genes in the ToppGene database as the background reference set and (ii) only including genetic markers of cortical cells as the background reference set ($N_{genes} = 5000$), including neurons, astrocytes, endothelial cells, microglia, oligodendrocytes[38] (see Supplementary Table 8 for the Bonferroni corrected p-values of the molecular functions, biological processes, and cellular processes listed in Fig. 3c). Complete gene ontology lists without and with background gene sets can be found on GitHub: https://github.com/VPNL/babies_graymatter/.

**Reporting summary**. Further information on research design is available in the Nature Research Reporting Summary linked to this article.

## Data availability

Source data[92] used in the analyses and to reproduce figures have been made freely available in Zenodo under accession code: https://doi.org/10.5281/zenodo.5514324 and are also provided as a Supplementary Data 1. Visualizations of quantitative $T_1$ surface maps generated for each subject are provided in the Supplementary Information file. Requests for further information or raw data should be directed to the Corresponding Author, Vaidehi S. Natu (vnatu@stanford.edu).

## Code availability

Quantitative T1 maps and anatomicals were generated using the mrQ software package (https://github.com/mezera/mrQ). DMRI data were generated and analyzed using mrTrix3 (https://github.com/MRtrix3/mrtrix3), mrDiffusion toolbox (http://github.com/vistalab/vistasoft) and FSL's top-up tool (https://fsl.fmrib.ox.ac.uk/). All custom code to produce figures and graphs are available on GitHub (https://github.com/VPNL/babies_graymatter). Custom codes were written in MATLAB 2017b.

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

## Acknowledgements

This research was supported by the Wu Tsai Neurosciences Institute Big Ideas Grant Phase I, Stanford University (Grill-Spector) and NIH R21 EY030588 (Grill-Spector). We would like to acknowledge Amy Kang, KK Barrows, Javier Marquis Lopez, Laura Villalobos, Lois Williams, and Alex Rezai for their contributions with gray and white matter segmentations of the infant brains and Caitlyn Estrada for her contribution to data collection. We would also like to thank Jiyeong Ha for her contributions towards the quality assurance check on the delineations of brain regions on the infant and adult brains. Additionally, we would like to thank Fiorella Carla Grandi for feedback and advice on the gene analyses.

## Author contributions

V.S.N. data preprocessing, statistical analysis, manuscript writing; M.R. study design, data acquisition, data preprocessing, statistical analysis, manuscript writing; H.W. sequence development; F.R.Q. participant recruitment, data acquisition, data preprocessing; H.K. participant recruitment, data acquisition, data preprocessing; N.L.A. data collection; M.G. data preprocessing (dMRI), S.B. data preprocessing (generation of T1-weighted images), A.A.M. sequence development, K.G.S. designed the study, oversaw all components of the study, and wrote the manuscript. All co-authors read and approved the submitted manuscript.

## Competing interests

The authors declare no competing interests.
