## [Transparent Peer Review File · Communications Biology]

Reviewers' comments:

Reviewer #1 (Remarks to the Author):

The authors report the results from an impressive data set obtained by a dense longitudinal sampling and hence allowing for a better understanding of the normative developmental trajectories using within-individual changes. The authors report a decrease of T1 and MD with age; they then show that the most expressed genes postnatally in visual cortex are those coding for myelin; they also show an asynchrony of maturation revealed in the two ways: 1) hierarchically lower regions of the two visual streams are more mature at birth; 2) less mature (hierarchically higher) regions experience a quicker maturation rate postnatally.

Methodologically the paper is very sound; I don't have any particular comments on the statistical or data preprocessing choices; they appear to be well justified.

The contribution to theory of the development of the visual system is also very clear.

The main ambition of the paper, however, as it seems, is to contribute to an understanding of general principles of brain development using visual system as a model. And I think a reader may remain wondering what exactly represents a conceptual breakthrough in this paper. The findings, even if some of them may be novel, are not really surprising to uncover novel organizational principles in the brain development; say, myelin growth postnatally is a known fact (including some myelin-mapping MRI research on early myelin development in white matter if not in cortex); asynchronous maturation (or ageing) is also a mainstream finding (one of the opening sentences reads "Primary sensory-motor cortices are more developed in infants than the prefrontal cortex¹⁻⁷, which is involved in complex cognitive functions", which with a support of 7 references suggests hierarchical asynchrony at the whole brain level). Findings that genes coding for myelin growth are expressed postnatally is probably very novel but could be well expected.

An interesting attempt was made to differentiate between growth vs. pruning hypotheses. However, this may not be easy given a complexity of factors influencing T1. A couple of alternative interpretations for the observed decrease could go as follows: 1) pruning could be associated with an increased concentration of iron-rich phagocytes and it is well known that iron contributes significantly to T1 (=decrease of T1), especially in cortex. (The authors draw a single reference that indicates that there is no iron in the neonatal brain, but this is an isolated study that looks specifically at a particular type of iron and is not generalisable enough). 2) a change in signal due to pruning may be overflown by the changes in myelination, especially considering that the neurites that got pruned may be unmyelinated and hence are less "visible" to MRI. I think the authors tacitly acknowledge this when talking about relative effects in: "However, there is an intense debate regarding the relative effects of microstructural growth and pruning and if they vary across cortical regions".

Further comments:

Figure 1D & F- the results could be more informative for the reader if time points of each subject were connected.

"Surprisingly, the change in R1 shows an opposite pattern, being progressively larger in later than earlier visual areas of both visual streams" – I think the interpretation needs to be more nuanced, as this is not exactly accurate, (the rate-of-change is not largest for the hierarchically highest regions, especially in dorsal stream (Fig 4B & D)).

Possibly use metric system for weights and length as it is more comprehensive for the readers outside

the US and UK?

Quality check on the delineation of regions of interest (ROIs). The authors selected V1 for their quality check of alignment, which is the most mature, adult-like, region. Would quality of alignment be similar for the later-developing regions?

"Intriguingly, the top most differentially expressed gene in primary sensory-motor and visual cortices is myelin basic protein (MBP), a gene associated with myelin generation and myelin sheath wrapping" – is this specific for visual areas?

There are a couple of cases when the meaning was a bit unclear:

"To work with magnitude images, we took the absolute value of the above signal equation and used it as the fitting model." (i.e., can a "value" be a "model"?)

"We reasoned that if regions identified from adult atlases would show better correspondence to manually defined regions in adults than infants, it would indicate that these atlases are in optimal for infants." (what does "in optimal" mean?)

Reviewer #3 (Remarks to the Author):

Natu and colleagues analyzed age dependency of tissue properties in cortical gray matter among infants, with particular focus on the hierarchy of visual areas. To this end, they acquired multi-contrast MRI dataset from 13 infants with 0, 3, and 6 months, and analyzed mean diffusivity (MD), quantitative T1 (qT1) or R1 in each cortical area. Wang atlas was used to identify an approximate location of visual areas. They found that qT1 in cortical gray matter becomes shorter during the development. They also report the evidence on hierarchical development of visual system, by demonstrating that (1) early visual cortex matures earlier than high-level visual cortex at birth and (2) high-level visual cortex matures with faster speed after a birth. Furthermore, they also analyzed the publicly available gene dataset to investigate biological cause of such change.

I admire author's extensive efforts on performing challenging data acquisition from infants and developing analysis pipelines on infant's data. I found that the data reported in this paper is very precious and finding on the relationship between development and hierarchy of visual areas is interesting one. The other strong point of the study is the use of the latest quantitative MRI methods, which are usually difficult to apply for infants due to longer acquisition time. Overall, I am enthusiastic on the author's approach and support a publication of this work. I believe that manuscript has some rooms for improvement, by clarifying following points.

Major points:

1. Authors repeatedly use a term "prevailing" (view/theory/hypothesis) and expressed the result in this paper will lead us to revisit such existing view (in Abstract, Introduction and Discussion). I am slightly confused on these statements, since second paragraphs of Introduction states that theories on microstructural development have been debated but later authors state that there is a "prevailing" theory. I found that these statements on previous literatures ("debated topic" vs. "prevailing theory") are difficult to follow. I am also not sure how much this theory is truly "prevailing" or not. As far as I saw it, Huttenlocher's paper (1997, J Comp Neurol), which has been cited in Discussion, only claimed the heterogeneity in synaptogenesis between auditory and prefrontal cortex and did not state any theories generalizable to striate and extrastriate cortex. I think that this manuscript can be more

logically coherent if authors omit the statement of “prevailing” theory, unless authors have a strong reason that there is an established theory in microstructural development and cortical hierarchy.

2. Authors analyzed the data using linear mixed model. A key assumption is that age dependency of diffusion or quantitative MRI measurements is linear. I’m not sure whether this is always true or not, since some studies reported that age dependency of MRI measurements in infant’s brain can be nonlinear (see Wu et al., 2020 Brain Structure and Function, 225, 2431-2445). While I feel a linear model seems to be appropriate by looking at the plot, I think that it will be helpful if authors could clearly justify the reasons on why they used a linear model in this case.

3. Are there any differences in hierarchical development between dorsal and ventral areas? By looking at Supplementary Figure 6, I thought that slopes might differ between dorsal and ventral areas, but it is hard to tell without statistical information. If there were statistically meaningful differences between dorsal and ventral areas, I think that they are worth reporting.

4. Change rate of R1 and MD (Figure 4B, 4D, S8F, S8H) looks like inverted U-shape. Are there any possible explanations for why they are not monotonic increase?

5. I think that an interpretive limitation of this study is that it is not practically possible to perform laminar analysis on this dataset due to a limit of spatial resolution. For example, the V1 is not typical in a sense of myeloarchitecture since it contains the stria of Gennari and shorter T1 in V1 at birth could be explained by an early maturation of Gennari. It is not fully clear whether the pattern of the development can be generalizable across different layers in V1. I think that it will be worthwhile to discuss this point, together with possible future direction in Discussion.

6. I think that discussion can be further extended, by discussing functional significance of findings. I believe that several vision science groups have measured development of visual functions, such as face recognition, contrast sensitivity or motion perception, in infants. While I’m not the expert of developmental vision science, I think that it will be interesting to discuss how maturation pattern of dorsal and ventral stream may or may not explain the development of visual functions investigated by developmental vision science studies.

7. Authors identified genes related to postnatal development of sensory cortices. They then estimated the enriched physiological processes. From the next sub-chapter (“Early visual areas are more myelinated at birth but myelinate at slower than later visual areas”), author use relatively strong statement on myelin for interpreting MRI data. I think that there is some logical gap here. The enrichment analysis showed multiple physiological processes, including not only myelin-related process, but also synapse signaling, axonal projections and dendritic spines. I thought that there remains a possibility that processes other than myelination can also account for developmental change of qT1 and MD data. If authors have strong reasons that non-myelin factors identified by gene analysis will not explain variance of MD and qT1, please clarify. If authors consider that they could not fully exclude other factors, I will suggest toning down arguments on myelin in later part of Results and Discussion.

Specific points:

p.3: “E.g., at 3 months,” may be “For example, at 3 months,”

p.7: “mean diffusivity” has been already defined as MD in prior sections. I think that authors can simply use “MD”.

p.9: Authors wrote “We find no evidence for cortical tissue pruning in sensory-motor cortices from 0 to 6 months of age”.

I’m not sure whether this argument can be supported by the data. A key assumption of the argument that qT1 and/or MD is sensitive to identify cortical tissue pruning. If authors have confidence of this argument, please include explanations on why qT1 and MD must be sensitive for such tissue change.

p.10: Authors used two different magnets for acquiring the data. It will be helpful to cite some references showing that qT1 may be relatively stable across different magnets, to justify this data collection strategy.

p.14: Some sentences of “dMRI” section misses period (“.”). Please also clarify whether authors used DTI or DKI to estimate MD.

Hiromasa Takemura (signed)

We thank the editor and the three reviewers for their thoughtful and detailed comments on our paper. We also appreciate the opportunity to address their comments and to revise and resubmit our manuscript. Below, we address and alleviate all the remaining comments raised by the reviewers. We believe that the changes implemented in the revision improved the manuscript, and that it is now suitable for publication.

Reviewer comments are in black; our responses are in blue.

Reviewers' comments:

Reviewer #1 (Remarks to the Author):

The authors report the results from an impressive data set obtained by a dense longitudinal sampling and hence allowing for a better understanding of the normative developmental trajectories using within-individual changes. The authors report a decrease of T1 and MD with age; they then show that the most expressed genes postnatally in visual cortex are those coding for myelin; they also show an asynchrony of maturation revealed in the two ways: 1) hierarchically lower regions of the two visual streams are more mature at birth; 2) less mature (hierarchically higher) regions experience a quicker maturation rate postnatally.

Methodologically the paper is very sound; I don't have any particular comments on the statistical or data preprocessing choices; they appear to be well justified.

The contribution to theory of the development of the visual system is also very clear.

We thank the reviewer for their enthusiastic summary of our study, for highlighting the clarity of our findings, noting the soundness of our methods and analyses, as well as for underscoring that our study is a worthy contribution to the field.

The main ambition of the paper, however, as it seems, is to contribute to an understanding of general principles of brain development using visual system as a model. And I think a reader may remain wondering what exactly represents a conceptual breakthrough in this paper. The findings, even if some of them may be novel, are not really surprising to uncover novel organizational principles in the brain development; say, myelin growth postnatally is a known fact (including some myelin-mapping MRI research on early myelin development in white matter if not in cortex); asynchronous maturation (or ageing) is also a mainstream finding (one of the opening sentences reads "Primary sensory-motor cortices are more developed in infants than the prefrontal cortex¹⁻⁷, which is involved in complex cognitive functions", which with a support of 7 references suggests hierarchical asynchrony at the whole brain level). Findings that genes coding for myelin growth are expressed postnatally is probably very novel but could be well expected.

Response 1: In response to the reviewer's comment asking us to better explain the conceptual breakthrough provided by our work, we have revised and extended the introduction to better situate our study relative to prior knowledge and highlight the gaps in knowledge. We also underscore that the reviewers' expectations from prior research - such as the presence of genes coding for myelin in the infant's cortex - are hypotheses that require empirical testing in humans rather than well known facts. We believe that to have an important conceptual advancement, it is necessary to conduct research to test theoretical predictions and either validate or falsify them, which we have done in our present research. We also now underscore that prior qMRI and dMRI studies in humans have largely examined white matter development that is indeed expected to myelinate throughout human development, but there is no prior knowledge on microstructural development of sensory-motor cortices in infancy.

In the Introduction, on page 2 we now write: "*Development of the cortical neuroarchitecture during early infancy is critical for the maturation of key sensory and cognitive functions and has lifelong consequences. During the first 6 months of life is when infants acquire crucial sensory-motor capacities¹*"

such as color, contrast, and spatial sensitivity² in the visual domain, and ability to lift the head, roll, grasp, and sit in the motor domain³. However, we have little knowledge of the rate, sequence, and microstructural mechanisms of development of human sensory-motor cortices that support these basic human abilities.

Present understanding of microstructural development in cortex is gleaned from histological investigations of a handful of sensory-motor and prefrontal regions in non-human primates⁴⁻⁹ and humans^{10,11}. Prior studies suggest that in infants, primary sensory-motor cortices are more developed than prefrontal cortex¹⁰⁻¹⁶, which is involved in complex cognitive functions. Additionally, histological research suggests that while cortex proliferates in infancy by growing synapses^{5,7,11}, dendrites^{6,8}, axons^{6,8}, and myelin¹⁰, it also prunes irrelevant connections and synapses^{6,8,11,17,18}. However, there is an intense debate regarding the relative effects of microstructural growth and pruning and if they vary across cortical regions.

Critically, generalization of developmental findings from non-human primates to humans is tenuous as human development is longer than other species and human visual cortex contains additional areas¹⁹ as well as additional gyri and sulci than non-hominid primate species²⁰. Moreover, as different cortical areas have unique cyto- and myelo-architecture^{21,22} for example, primary visual cortex - V1 - has a unique cytoarchitecture (Stria of Gennari) already at birth¹², to achieve significant advancement in understanding the development of sensory systems in humans it is necessary to study the development of multiple brain areas within a human cortical system.

To fill these glaring gaps in knowledge, we leveraged advancements in quantitative magnetic resonance imaging (qMRI)²³⁻²⁵ and diffusion MRI (dMRI)²⁶ to develop in vivo methodologies that are optimized for the infant brain. Up till now, only a handful of studies have used these methods to examine brain development^{†14-16,27} focusing predominantly on white matter development^{†15,28-30}."

An interesting attempt was made to differentiate between growth vs. pruning hypotheses. However, this may not be easy given a complexity of factors influencing T1. A couple of alternative interpretations for the observed decrease could go as follows: 1) pruning could be associated with an increased concentration of iron-rich phagocytes and it is well known that iron contributes significantly to T1 (=decrease of T1), especially in cortex. (The authors draw a single reference that indicates that there is no iron in the neonatal brain, but this is an isolated study that looks specifically at a particular type of iron and is not generalizable enough). 2) a change in signal due to pruning may be overflowed by the changes in myelination, especially considering that the neurites that got pruned may be unmyelinated and hence are less "visible" to MRI. I think the authors tacitly acknowledge this when talking about relative effects in: "However, there is an intense debate regarding the relative effects of microstructural growth and pruning and if they vary across cortical regions".

Response 2: We agree with the reviewer that due to the complexity of factors that influence quantitative T₁ / R₁ metrics in cortex, it is indeed challenging to optimally deduce the mechanisms that differentiate between tissue growth or tissue pruning from *in vivo* metrics alone. Although we find no evidence for increases in T₁ during early infancy, which would suggest tissue is pruned, it is possible that pruning might still occur in cortical tissue, but its effects on T₁ are smaller than microstructural growth effects, which dominate MRI metrics. Additionally, as the reviewer suggested there might be also other interpretations of our MR findings, which we now include in the discussion as follows.

On page 11 under sub-heading "Cortical microstructure grows exuberantly during early human infancy" we write: "Our data show strong effects of cortical increases in R₁ and decreases in MD supporting microstructural tissue growth in sensory-motor cortices from 0 to 6 months of age. These findings: (i) are supported by our transcriptomic analyses revealing myelination, synaptogenesis, and dendritic processes as key mechanisms of infants' sensory-motor cortex development, and (ii) are consistent with prior histological data showing synaptogenesis¹¹ and dendritic growth⁶ in primate V1 and A1 during early

infancy. While we do not find evidence for pruning, due to the complexity of factors that impact R_1 (T_1) in cortex, we acknowledge that we cannot conclude from qMRI metrics alone that there is no cortical pruning during early infancy. For example, it is possible that pruning effects may not be visible to qMRI metrics if the pruned neurites are unmyelinated. Additionally, changes in iron due to phagocytosis associated with pruning⁴³, may modulate R_1 (T_1), but may be obscured by larger effects of cortical myelination. Finally, our results do not preclude the possibility that pruning may occur later in infant development^{9,11} following this exuberant microstructural tissue growth during the first 6 months of life. However, our transcriptomic data provide complementary evidence for growth of multiple cortical tissue components during early infancy. Future histological investigations in pediatric samples containing sensory-motor cortices can elucidate precisely which cellular components develop during infancy in each visual area.”

Further comments:

Figure 1D & F– the results could be more informative for the reader if time points of each subject were connected.

Response 3: Thank you for this suggestion. We have updated **Supplemental Figures S3 and S4**, which now show the lines connecting T_1 and MD measures across time points per individual for the four primary sensory regions (V1, M1, S1, and A1, left and right hemispheres) related to **Figs. 1D and F**. Each participant is depicted with a different symbol. Graphs show decreasing T_1 and MD trends in the four primary sensory areas across each participant.

We report this in the Results section on page 3, we write: “Across all primary sensory-motor regions, mean T_1 substantially decreased from $2.03s \pm 0.07s$ (mean \pm standard deviation (SD)) in newborns, to $1.87s \pm 0.08s$ at 3 months, to $1.74s \pm 0.06s$ at 6 months. These reductions were observed in each individual infant across time points (**Supplementary Fig. S3A**). Analysis of MD in these areas revealed similar significant, linear decreases from 0 to 6 months (LMM slopes, rate of MD change: -9.36×10^{-7} to -1.01×10^{-6} [$\text{mm}^2/\text{s}/\text{day}$], $P_s < 0.001$, **Supplementary Table S3**, all stats; right hemisphere: **Figs. 1E, 1F**, left hemisphere: **Supplementary Fig. S4**), which are evident in individual participants (**Supplementary Fig. S4A-B**). “

“Surprisingly, the change in R_1 shows an opposite pattern, being progressively larger in later than earlier visual areas of both visual streams” – I think the interpretation needs to be more nuanced, as this is not exactly accurate, (the rate-of-change is not largest for the hierarchically highest regions, especially in dorsal stream (Fig 4B & D)).

Response 4: We agree with the reviewer that our original description of the results in **Fig 4B and D** was oversimplified. We have modified the text to provide a more nuanced description of the results.

On page 9, we now write: “Changes in R_1 during the first 6 months become progressively larger from V1 (right V1d: 0.59 ± 0.06 [$\text{ms}^{-1}/\text{day}$]) to IPS0 in the dorsal stream (right IPS0: 0.76 ± 0.04 [$\text{ms}^{-1}/\text{day}$], **Fig. 4B**) and from V1 (right V1v: 0.55 ± 0.06 [$\text{ms}^{-1}/\text{day}$]) to VO2 in the ventral stream (right of 0.76 ± 0.04 [$\text{ms}^{-1}/\text{day}$], **Fig. 4D**, **Supplementary Table S9**). While R_1 in IPS0 and VO2 develop $\sim 28\%$ faster than in V1 during the first 6 months of life, the highest retinotopic areas in both streams showed a lower rate of R_1 change. That is, the rate of change in R_1 in IPS2/3 is similar to that of V2d (**Fig. 4B**) and that of PHC1/2 is similar to V3v (**Fig. 4D**). “

Possibly use metric system for weights and length as it is more comprehensive for the readers outside the US and UK?

Response 5: Thank you for this suggestion. We now report the weights and length of the infants in metric units for readers outside the United States and United Kingdom.

On page 14, under “Expectant mother and infant screening procedure”, we write: “Exclusionary criteria for infants were preterm birth (< 37 gestational weeks), low birthweight (< 2.49 kgs), small height (< 45 cms), any congenital, genetic, or neurological disorders, visual problems, complications during birth that involved the infant (e.g., NICU stay), history of head trauma, and contraindications for MRI (e.g., metal implants).”

Quality check on the delineation of regions of interest (ROIs). The authors selected V1 for their quality check of alignment, which is the most mature, adult-like, region. Would quality of alignment be similar for the later-developing regions?

Response 6: We chose the quality check on the delineation of the calcarine sulcus as the benchmark as the location of V1 in the calcarine is well documented in adults and it can be identified anatomically in each individual brain. Although V1 is relatively more mature than the other regions in the visual hierarchy, it is not yet fully adult-like and undergoes significant microstructural changes during the first 6 months of life as we report here. Furthermore, finding that there were no significant age-related differences in alignments between infants and adults suggests that cortex-based alignment of brain atlases based on adult templates to infants’ cortical surfaces is similar in quality to this transformation in adults.

An important question that the reviewer raises is whether this structural-functional correspondence extends to higher-level visual areas that are less mature microstructurally at birth. To address this question, we added a new analysis examining the quality of alignment of two later-developing areas that are higher in the ventral hierarchy: retinotopic parahippocampal areas PHC-1 and PHC-2. We chose these areas for two reasons: (i) in adults, they are anatomically consistent: PHC-1 and 2 are located anterior to VO-2 extending across the collateral sulcus (CoS) into the posterior parahippocampal cortex (PHC) (Arcaro et al., 2009; *Journal of Neuroscience*), and (ii) in adults, they overlap with the functionally defined parahippocampal place area (PPA, also referred to as CoS-places, Weiner et al., 2018, *Neuroimage*).

Results shown in **new Supplemental Fig. S9** demonstrate a similar quality of cortex-based alignment of PHC-1, PHC-2, and CoS-places in infants as we found for V1. **Supplemental Fig S9** shows the results of cortex-based alignment of PHC-1 and PHC-2 from the Wang atlas (Wang et al., 2015, *Cerebral Cortex*), the anatomically defined collateral sulcus (CoS) from the Desikan atlas (Desikan et al., 2006; *Neuroimage*) and the place-selective region (Cos-places) from the Rosenke atlas (Rosenke et al., 2021, *Cerebral Cortex*) in 5 sample infants at 3 timepoints: newborn, 3 and 6 months. Results demonstrate that (I) the CoS from the Desikan atlas is correctly aligned to each infant’s CoS, (ii) PHC-1 and 2 are accurately aligned to the CoS, and (iii) CoS-places overlaps PHC-1 and 2 in infants as found in adults. Out of the 60 hemispheres, we found a single hemisphere where the anatomical CoS was shifted more laterally and less aligned to the infant’s CoS. However, even in this hemisphere, PHC-1 and 2 and CoS-Places were still aligned to each other and the CoS. This indicates there is a consistent anatomical alignment and a stable functional-structural relationship when mapping from adult atlases to infant brains using cortical-based alignment.

We also added this analysis to the Methods on page 19, where we write: “As V1 is relatively more developed than other areas of the cortex in newborns, we also examined the quality of cortex-based alignment in infants of two later-developing regions, parahippocampal areas PHC-1 and PHC-2. In adults, these regions are (i) located anterior to VO-2 and extend across the collateral sulcus (CoS) into the posterior parahippocampal cortex (PHC)⁸⁷, and (ii) overlap with the functionally defined parahippocampal place area⁸⁷ (also referred to as CoS-places⁸⁸). We tested if cortex-based alignment of these high-level

ROIs to the infant brain will show the same structural-anatomical coupling in infants as in adults. To do so, we used cortex-based alignment to project from the FreeSurfer average cortical surface to each infant's cortical surface at each timepoint: (i) the CoS from the Desikan atlas⁷⁸, (ii) PHC-1 and 2 from the Wang atlas³⁴, and (iii) CoS-places from the Rosenke atlas⁷⁹. Results shown in **Supplemental Fig. S9** demonstrate similar functional-structural coupling of PHC-1 and 2 to the CoS as well as similar coupling between PHC-1 and 2 to the CoS-places in infants. Out of the 60 hemispheres, there was a single hemisphere where the anatomical CoS was shifted more laterally and was less aligned to the infant's CoS. However, even in this hemisphere, PHC-1 and 2 and CoS-Places were still aligned to each other and the CoS. This analysis validates our delineation of ROIs in infants using cortex-based alignment of data from adult atlases."

"Intriguingly, the top-most differentially expressed gene in primary sensory-motor and visual cortices is myelin basic protein (MBP), a gene associated with myelin generation and myelin sheath wrapping" – is this specific for visual areas?

Response 7: When examining the differentially expressed genes for the postnatal versus prenatal tissue samples including visual regions (primary visual area V1, parietal and temporal expanses overlapping visual regions of the dorsal and ventral visual streams) as well as other primary sensory-motor cortices (M1, S1, and A1) to match our *in vivo* data we found that MBP was the most expressed gene. The reviewer asks interesting question that is if the expression of MBP also occurs specifically for visual areas.

To examine this question, we added a new analysis examining the differential expression of genes postnatally vs. prenatally using only the samples that contained visual areas (V1, parietal and temporal expanses overlapping visual regions of the dorsal and ventral visual streams). This differential analysis generated a list of several thousand genes that are expressed significantly more in visual cortical expanses postnatally than prenatally. To determine the most differentially expressed genes, we selected the genes with the largest expression fold changes ($FC > 4$) and assessed their significance after Bonferroni correction for multiple comparisons ($P < 5.7 \times 10^{-6}$). Ninety genes survived these criteria. Results show that the top-most differentially expressed gene is myelin basic protein (MBP). Several other myelin-related genes, including myelin-associated oligodendrocytic basic protein (MOBP), myelin-associated glycoprotein (MAG), and proteolipid protein (PLP-1), are also among the top 20 most expressed genes in visual cortex postnatally. To elucidate the molecular and cellular pathways linked to the genes, we used the ToppGene toolbox (<https://toppgene.cchmc.org>) to map this list of expressed genes to the enriched physiological processes. As in our main findings, ToppGene reported significant enrichment of several biological processes related to structural constituents of myelin sheath ($P_{\text{Bonferroni Corrected (BC)}} = 1.75 \times 10^{-3}$), synaptic signaling ($P_{\text{BC}} = 1.57 \times 10^{-17}$), and cellular components of dendritic trees and spines ($P_{\text{BC}} = 1.24 \times 10^{-3}$).

We report these data in the manuscript on page 8 of the Results, where we write: "*To test if the expression of MBP also occurs specifically for visual areas, we conducted an additional analysis of the differential expression of genes postnatally vs. prenatally using only the samples that contained visual areas (V1, parietal and temporal expanses overlapping visual regions of the dorsal and ventral visual streams). Results showed that the top-most differentially expressed gene in visual cortex postnatally is MBP. Several other myelin-related genes, including MOBP, MAG, and PLP-1, are also among the top 20 most expressed genes in visual cortex postnatally. As in our main findings, ToppGene also reported significant enrichment of several biological processes related to structural constituents of myelin sheath ($P_{\text{BC}} = 1.75 \times 10^{-3}$), synaptic signaling ($P_{\text{BC}} = 1.57 \times 10^{-17}$), and cellular components of dendritic trees and spines ($P_{\text{BC}} = 1.24 \times 10^{-3}$).*"

There are a couple of cases when the meaning was a bit unclear:

"To work with magnitude images, we took the absolute value of the above signal equation and used it as the fitting model." (i.e., can a "value" be a "model"?)

Response 8: Thanks for this comment. In response, we have clarified our methods on quantitative T_1 relaxation time modeling. On page 16 we write: *"We applied an absolute value operation on both sides of the equation and used the resulting equation as the fitting model. We use the absolute value of the signal equation because we use the magnitude images to fit the model. The magnitude images only keep the information about the strength of the signal but not the phase or the sign of the signal."*

"We reasoned that if regions identified from adult atlases would show better correspondence to manually defined regions in adults than infants, it would indicate that these atlases are in optimal for infants." (what does "in optimal" mean?)

Response 9: Thanks for the comment. We have clarified our description of quality checks on the delineation of regions of interest (ROIs).

On page 19 we write: *"Typically for functional and anatomical analysis in adults, brain atlases generated using average adult brains are used for delineating regions of interest using cortex-based alignment to the cortical surface of each individual subject. For instance, when using the Wang atlas³⁴ to delineate primary visual area, V1, which lies in the calcarine sulcus, we expect that cortical based alignment of V1 from the Wang atlas to the individual brain should map V1 accurately to the calcarine sulcus of the individual cortical surface. As we used cortex-based alignment and atlases developed for the adult brain to define areas in the infant brain, we first sought to test how well anatomical ROIs from adult atlases were mapped to infants' brains. To test this correspondence, we compared how the calcarine sulcus manually defined on each individual brain compares to the cortex based aligned calcarine sulcus from the Desikan atlas⁷⁸, an anatomical parcellation of the brain, in the average adult FreeSurfer cortical surface⁸⁴."*

Reviewer #2 (Remarks to the Author):

This paper harnesses the quantitative aspect of anatomical, quantitative MRI, and diffusion MRI to shed light onto cortical development in infants. Specifically they test the hypotheses that 1) infant's cortex undergo microstructural growth and myelination; 2) there is evidence of a pruning process.

The article is clearly written, results are well presented, seem to be sound, and the analyses are as clean and simple which is a big plus in bringing to light the studied phenomena. Furthermore, the combination of imaging data with genetics provides interesting ancillary information in terms the specificity of the microstructural growth process in infants as the results were correlated with gene expressions related to myelin, as well as cellular components of axonal projections and dendritic spines. I find particularly interesting the hierarchical developmental pattern that the authors find across different parieto-occipital and occipito-temporal streams. The results are interesting and clearly worth being published.

We thank the reviewer for their enthusiastic summary of our study, for noting the clarity of our methods and analyses, and highlighting that our findings are interesting and worthy of publication.

My main criticism to this manuscript is with respect to the writing, specifically the discussion of the results. I find the discussion too succinct. Would it be possible for the authors to discuss animal studies which show similar patterns? The study such myelination or microstructural patterns in non-human primates and other mammals has been long studied (for instance in the ferret, Jackson et al 1989; or pruning studies of

LaMantia et al 1994). Positioning with respect to the work of C. Lebel on ageing in the brain analysed through diffusion MRI in general could also be interesting.

Response 10: We agree with the reviewer that the original discussion was succinct due to word limit constraints. We have now expanded our discussion to relate our findings to previous animal work that shows similar developmental patterns as well as elaborate on the work of C. Lebel on ageing in the brain.

On page 11 we write: *“Gene analyses reveal that MBP is the top-most differentially expressed gene after birth, and both ex vivo and in vivo studies show that cortical R_1 increases with higher myelin content^{24,32}. Consequently, this suggests that myelination contributes to tissue changes in infants’ cortex measured with R_1 . These findings are transformative for three reasons. First, they uncover the development of cellular and biological mechanisms in infant sensory motor cortices. While research in other species also revealed postnatal myelin growth and synaptogenesis^{7,10}, as well as initial overproduction of axons^{8,44}, generalizing across species is challenging as species vary significantly in brain size, development trajectory⁴⁵, and cortical functional organization. Hence, to uncover the development of the human brain, it is critical to study the trajectory of biological mechanisms in humans as well as between brain regions across the human life-span⁴⁶.*

Second, our findings link specific biological mechanisms to in vivo MRI measurements. Moreover, microstructural development likely precedes or occurs together with functional development, giving us an opportunity to track the earliest developments in cortex. This is important as it lays a necessary foundation for developing in vivo markers for diagnosing typical and atypical brain development. As babies in the first year of life are a highly vulnerable population, the ability to use in vivo tools to diagnose neurodevelopmental disorders and deficiencies as well as intervene early could not be of greater importance.

Third, our findings suggest that not only synaptogenesis^{5,11,47} and neurite sprouting⁶ but also cortical myelination is critical for the development of brain function and ultimately behavior. As cortical myelination is thought to be activity dependent^{48–50}, these data further suggest that increases in cortical R_1 during early infancy may be linked to functional changes in the same cortical areas. This opens exciting new avenues for future research in infants that will examine the link between cortical myelination and functional brain development and suggests that cortical myelination should be considered in any future infant research^{51–53}.”

A second criticism is regarding the statistical protocols. The authors vary between Bonferroni and FDR correction strategies for their analyses. What has motivated this heterogeneity? The sensitivity of the experiments?

Response 11: We thank the reviewer for their comment. Different toolboxes use different default statistics. To address this comment, and to have a consistent statistical method for correction for multiple comparisons, in the revised manuscript we only use Bonferroni correction for all our analyses as this is a more conservative method. Specifically, we have (i) updated **Fig. 3C**, which shows the gene enrichment analysis, to report Bonferroni corrected p-values for each of the molecular and biological processes, (ii) updated the corresponding statistics in **Supplementary Table S8**, and (iii) updated the Results section with the corresponding Bonferroni corrected p-values.

On page 8, we write: *“To further elucidate the molecular and cellular pathways linked to these 95 genes, we used the ToppGene toolbox (<https://toppgene.cchmc.org>)³⁷ to map this list of expressed genes to the enriched physiological processes. Comparing the 95 most significantly-expressed genes to all protein-coding genes as the background set, ToppGene reported significant enrichment of several biological processes related to: (i) myelination ($P_{\text{Bonferroni_corrected (BC)}} = 4.44 \times 10^{-3}$), (ii) structural constituents of myelin sheath ($P_{\text{BC}} = 2.05 \times 10^{-5}$), (iii) axonal ensheathing ($P_{\text{BC}} = 4.69 \times 10^{-3}$), (iv) synaptic signaling ($P_{\text{BC}} = 5.36 \times 10^{-12}$),*

and (v) cellular components of dendritic trees and spines ($P_{BC} = 3.33 \times 10^{-4}$) (Fig. 3C, Supplementary Table S8). These processes remained enriched in a control analysis in which we compared this list of top-95 most expressed genes to a different background gene set that was restricted to markers of cortical cells (neurons, astrocytes, endothelial cells, microglia, oligodendrocytes³⁸).“

In terms of minor criticisms

* The fact that T1 and MD are in physical units does not make them instantaneously a measurement which is comparable across subjects and scanners which is one of the main points of the current trend of quantitative MRI, the authors might want to tone down their write-up of these characteristics

Response 12: The reviewer comments that although T₁ and MD are in physical units they are not instantaneously comparable across scanners and subjects. Nonetheless, several studies have shown the importance of these metrics being in physical units and that they are in fact replicable across scanners and comparable across subjects, thus enabling examination of life-span brain development (Yeatman et al 2014; Nature Communications). In fact, only a small variability in quantitative *in vivo* metrics has been found across different sites using the same MRI system type (Weiskopf et al., 2013; *Frontiers in Neuroscience*; Gracien et al., 2020; *Neuroimage*). Further, both Weiskopf et al. 2013, and Gracien et al., 2020 show that the discrepancies between qMRI data acquired with different scanner models are low; Biases and variance in metrics are also low if identical acquisition sequences and identical protocols are used for data collection. In our study we used the same scanner model (two identical 3T GE Discovery MR750 Scanners (GE Healthcare) and Nova 32-ch head coils (Nova Medical)), as well as same protocols and sequences across the two scanners at Stanford. Consistent with prior work, in our data, the values across subjects at a particular timepoints are consistent across the two scanners.

On page 15 of the Methods section, we now write: “Data were acquired at two identical 3T GE Discovery MR750 Scanners (GE Healthcare) and Nova 32-ch head coils (Nova Medical) located at Stanford University: (i) Center for Cognitive and Neurobiological Imaging (CNI) and (ii) Lucas Imaging Center using identical acquisition sequences and protocols. Prior work has shown that quantitative imaging metrics are replicable across scanners and subjects and have low inter-site discrepancies^{68,69}.“

* T1 and MD are hardly innovative *in vivo* metrics. They have been used to assess myelin and microstructure configurations for at least 20 years, which in terms of the MRI field is quite long (see e.g., Wang et al Brain 2011)

Response 13: We agree with the reviewer that T₁ and MD have been used to study brain microstructure and myelin over the last 10 years and removed the words ‘innovative’ and ‘novel’ to describe quantitative and diffusion metrics in the Introduction and Discussion. However, these methods have been mainly used to assess development of white matter or sub-cortical structures (e.g., Deoni et al., 2011; Dubois et al., 2014; Lebel and Deoni, 2018) rather than cortical development, and the handful of studies that include cortical data (Lebenberg et al. 2019, Ball et al., 2013), have not used qMRI to determine longitudinal development of any cortical area and entire sensory-motors systems, in individual infants over time.

To address the reviewer’s comment, we revised the introduction and added citations, on page 2 we write: “To fill these glaring gaps in knowledge, we leveraged advancements in quantitative magnetic resonance imaging (qMRI)^{23–25} and diffusion MRI (dMRI)²⁶ to develop *in vivo* methodologies that are optimized for the infant brain. Up till now, only a handful of studies have used these methods to examine brain development^{†14–16,27} focusing predominantly on white matter development^{†15,28–30}. Quantitative measurements of proton relaxation time (T₁, which depends on the physiochemical tissue environment) from qMRI and mean diffusivity (MD, which depends on the density and structure of tissue through which water diffuses) from dMRI enable quantifying and longitudinally measuring the amount of brain tissue within a voxel (3D pixel in an MRI image, 1-2mm on a side) related to the neuropil³¹ and myelin³². Thus, these quantitative metrics provide noninvasive means to glean into microstructural changes as well as

disambiguate developmental hypotheses as T_1 and MD are lower in tissue with denser microstructure^{24,31,32}. We predicted that if cortical microstructure proliferates, T_1 and MD will decrease during infancy, but if microstructure is pruned, T_1 and MD will increase. We tested these hypotheses in (i) primary sensory-motor areas to relate our measurements to prior histological studies, and (ii) across visual areas spanning two processing hierarchies³³. This offers an exciting opportunity to investigate the sequence and rate of microstructural development across an entire cortical system with precision and fine granularity for the first time.”

* The last point is that I find the manuscript structure, easy to read, but disorienting. The authors have decided to split in an ever-so-subtle what the Results/Experiments section from the introduction, and incorporate some bits of discussion into it. It's good for a first reading but in terms of trying to re-read the paper several times to find specific bits of evidence or results in the paper is disorienting.

Response 14: We apologize for the disorientation between the Introduction and Results sections, and the lack of subheadings in the paper. We have now added separate section headings for Introduction, Results, and Discussion.

Reviewer #3 (Remarks to the Author):

Natu and colleagues analyzed age dependency of tissue properties in cortical gray matter among infants, with particular focus on the hierarchy of visual areas. To this end, they acquired multi-contrast MRI dataset from 13 infants with 0, 3, and 6 months, and analyzed mean diffusivity (MD), quantitative T1 (qT1) or R1 in each cortical area. Wang atlas was used to identify an approximate location of visual areas. They found that qT1 in cortical gray matter becomes shorter during the development. They also report the evidence on hierarchical development of visual system, by demonstrating that (1) early visual cortex matures earlier than high-level visual cortex at birth and (2) high-level visual cortex matures with faster speed after a birth. Furthermore, they also analyzed the publicly available gene dataset to investigate biological cause of such change.

I admire author's extensive efforts on performing challenging data acquisition from infants and developing analysis pipelines on infant's data. I found that the data reported in this paper is very precious and finding on the relationship between development and hierarchy of visual areas is interesting one. The other strong point of the study is the use of the latest quantitative MRI methods, which are usually difficult to apply for infants due to longer acquisition time. Overall, I am enthusiastic on the author's approach and support a publication of this work. I believe that manuscript has some rooms for improvement, by clarifying following points.

We thank the reviewer for their enthusiastic summary of our study and findings, for highlighting our efforts in data acquisition and developing infant data analysis pipeline, as well as for supporting the publication of our study.

Major points:

1. Authors repeatedly use a term “prevailing” (view/theory/hypothesis) and expressed the result in this paper will lead us to revisit such existing view (in Abstract, Introduction and Discussion). I am slightly confused on these statements, since second paragraphs of Introduction states that theories on microstructural development have been debated but later authors state that there is a “prevailing” theory. I found that these statements on previous literatures (“debated topic” vs. “prevailing theory”) are difficult to follow. I am also not sure how much this theory is truly “prevailing” or not. As far as I saw it, Huttenlocher's paper (1997, J Comp Neurol), which has been cited in Discussion, only claimed the heterogeneity in synaptogenesis between auditory and prefrontal cortex and did not state any theories generalizable to striate and extrastriate cortex. I think that this manuscript can be more

logically coherent if authors omit the statement of “prevailing” theory, unless authors have a strong reason that there is an established theory in microstructural development and cortical hierarchy.

Response 14: As per the reviewer’s request we have now removed the wording “prevailing theory” from the manuscript.

On page 1, we write: *“This overturns the prominent view that visual areas that are most mature at birth develop fastest.”*

2. Authors analyzed the data using linear mixed model. A key assumption is that age dependency of diffusion or quantitative MRI measurements is linear. I’m not sure whether this is always true or not, since some studies reported that age dependency of MRI measurements in infant’s brain can be nonlinear (see Wu et al., 2020 Brain Structure and Function, 225, 2431-2445). While I feel a linear model seems to be appropriate by looking at the plot, I think that it will be helpful if authors could clearly justify the reasons on why they used a linear model in this case.

Response 15: The reviewer asks for a justification for modeling the developmental trends with a linear mixed model. As we have only three time points (newborns, 3 months, and 6 months), and since the developmental trend appear linear (**Figs 1-2**), we used linear mixed models (LMM) to fit our data. To validate the goodness of the model, we now report the proportion variance explained by the LMM model (R^2 in **Supplementary Tables S2-S5** and **S9**). The linear trends in our data are also present in other studies of brain development during early infancy (e.g., Wu et al 2020). Nonetheless, we agree with the reviewer that if we had measured brain development over a longer period across the lifespan, a non-linear model may have provided better fits, as developmental effects may slow down or plateau at certain ages.

We now added to the Methods section a justification for the LMM, on page 20 we write: *“While we expect that developmental trends will eventually asymptote over the lifespan^{89,90}, during the first 6 months of life, our data showed linear trends (**Figs 1, 2**) and in general LMM produced good fits to the data (R^2 in **Supplementary Tables S2-S5** and **S9**).”*

3. Are there any differences in hierarchical development between dorsal and ventral areas? By looking at Supplementary Figure 6, I thought that slopes might differ between dorsal and ventral areas, but it is hard to tell without statistical information. If there were statistically meaningful differences between dorsal and ventral areas, I think that they are worth reporting.

Response 16: The reviewer asks if there are developmental differences between the dorsal and ventral visual streams. To test for developmental differences between the two streams, we added new linear mixed model analyses with factors age, ROIs, and stream to examine differences between ventral and dorsal streams in the three MR metrics T_1 , MD, and R_1 . In general, we find no significant differences between the dorsal and ventral streams in the linear mixed models using T_1 and R_1 . However, there is a small but significant difference in MD. See figure below, where we show mean T_1 , R_1 , and MD values across the streams (green dots: ventral stream, blue dots: dorsal stream, each time point: all visual areas of a single infant). Results show no significant developmental differences between the streams for the R_1 and T_1 measures (stats below), and a small difference in the first and 6th month of life in MD measurements across the streams. This effect is significant for MD suggesting that MD is more mature in the ventral stream at birth, but develops faster in dorsal stream during the first 6 months of life. Although, we find no consistent developmental differences with quantitative MR metrics between the dorsal and ventral visual streams in the first 6 months of life, it is possible that there might be developmental differences between these streams as regions along the two streams become functionally more developed and more specialized as children develop.

We now report these findings in the Results section on page 6 where we write: “To test if there are developmental differences between the dorsal and ventral streams, we fit another LMM across T_1 data of all visual areas spanning both streams with factors of age, area, and stream. T_1 significantly varied with age ($t_{1012} = 9.877$, $P = 4.95 \times 10^{-22}$) and area ($t_{1012} = 5.63$, $P = 2.38 \times 10^{-8}$), but there were no significant differences across streams ($t_{1012} = .89$, $P = .36$) and no interactions ($ts < 1.5$, $Ps > 0.12$). Together these data show that both birth values and development of T_1 are heterogenous across areas of visual processing streams.

Similar results were observed with MD: (i) MD significantly decreases from newborns to 6-month-olds, and (ii) estimated MD at birth systematically increases from V1 to later visual areas of each stream (**Supplementary Figs. S6, S7, Supplementary Table S5**). LMM of MD data across areas of a stream with factors of age, area, and hemisphere showed that in both streams, MD significantly decreased with age ($ts > 3.82$, $Ps > 1.50 \times 10^{-4}$) and did not significantly vary across hemispheres ($ts < 1.37$, $Ps > .17$). MD development varied significantly across visual areas of the dorsal stream ($t_{496} = 3.38$, $P = 7.71 \times 10^{-4}$); there was a nonsignificant trend in the ventral stream ($t_{440} = 1.74$, $P = 0.08$). Comparing MD development across the ventral and dorsal streams, shows significant MD development with age ($t_{944} = 10.30$, $P = 1.13 \times 10^{-23}$), with differential development across areas ($t_{944} = 6.04$, $P = 2.13 \times 10^{-9}$) and streams ($t_{944} = 2.18$, $P = 0.02$) as MD decreases more in the dorsal than ventral stream in early infancy.”

On page 10 we write: “To test differences between R_1 development across the two streams we ran another LMM on all visual areas with factors of age, area, and stream. Results reveal that R_1 significantly varied with age ($t_{1012} = 11.21$, $P = 1.30 \times 10^{-27}$) and area ($t_{1012} = 4.81$, $P = 1.65 \times 10^{-6}$), but there was no significant difference in R_1 development across streams ($t_{1012} = .82$, $P = 0.40$) or interactions ($ts < 1.64$, $Ps > 0.1$). These results suggest that R_1 at birth and R_1 development varies across visual areas of ventral and dorsal visual processing streams.”

On page 21 in the Methods section, we write: “Finally, we used a third set of the linear mixed models to test if there are any developmental differences in T_1 , R_1 , MD between the dorsal and ventral streams, with factors: age, area, and stream. In this case as well the random intercept/random slope model best fit the data (model comparison, $Ps < 0.01$). “

4. Change rate of R_1 and MD (Figure 4B, 4D, S8F, S8H) looks like inverted U-shape. Are there any possible explanations for why they are not monotonic increase?

Response 17: We agree with the reviewer that development rate of R_1 and MD increases up to a certain point in the hierarchy and subsequently decreases. We elaborate more about the shapes of the change in R_1 curves in the results.

On page 9 we write: “Changes in R_1 during the first 6 months become progressively larger from V1 (right V1d: 0.59 ± 0.06 [ms^{-1}/day]) to IPS0 in the dorsal stream (right IPS0: 0.76 ± 0.04 [ms^{-1}/day], **Fig. 4B**) and from V1 (right V1v: 0.55 ± 0.06 [ms^{-1}/day]) to VO2 in the ventral stream (right of 0.76 ± 0.04 [ms^{-1}/day], **Fig. 4D, Supplementary Table S9**). While R_1 in IPS0 and VO2 develop ~28% faster than in V1 during the first 6 months of life, the highest retinotopic areas in both streams showed a lower rate of R_1 change. That is, the rate of change in R_1 in IPS2/3 is similar to that of V2d (**Fig. 4B**) and that of PHC1/2 is similar to V3v (**Fig. 4D**). “

5. I think that an interpretive limitation of this study is that it is not practically possible to perform laminar analysis on this dataset due to a limit of spatial resolution. For example, the V1 is not typical in a sense of myeloarchitecture since it contains the stria of Gennari and shorter T1 in V1 at birth could be explained by an early maturation of Gennari. It is not fully clear whether the pattern of the development can be generalizable across different layers in V1. I think that it will be worthwhile to discuss this point, together with possible future direction in Discussion.

Response 18: We agree with the reviewer that an interpretive limitation of our study is that it is not practically possible to perform laminar analysis on this dataset due to limitations of spatial resolution (i.e., 2 mm) of the MR measurements. In fact, as the reviewer noted, laminar analysis showing how T1/ R_1 varies from pial to white matter would be highly informative for understanding how different cortical layers emerge at birth and how they develop during early infancy. We now discuss directions for future laminar analyses in the Discussion.

On page 12 we write: “Our data across the visual system reveal three important findings. First, contrary to the theory that V1 is myelinated at birth^{12,57}, our analyses of R_1 suggest that while V1 is more developed than other visual areas at birth, it continues to develop and profoundly myelinates during the first six months of life. An interesting question for future research is whether the earlier maturation of V1 is due to overall higher myelination across cortical layers or is specific to the earlier myelination of the Stria of Gennari, which is highly myelinated in adults⁵⁸. Laminar analyses in histological pediatric samples will be instrumental for answering this open question.”

6. I think that discussion can be further extended, by discussing functional significance of findings. I believe that several vision science groups have measured development of visual functions, such as face recognition, contrast sensitivity or motion perception, in infants. While I’m not the expert of developmental vision science, I think that it will be interesting to discuss how maturation pattern of dorsal and ventral stream may or may not explain the development of visual functions investigated by developmental vision science studies.

Response 19: The reviewer suggested to extend the discussion and link to the development of visual function. We note that any link to function development or behavior at this point is speculative as we have not measured visual function in our participants. Nonetheless, we expanded the Discussion to relate the impact of our findings on understanding visual function in future research.

On page 13 we write: “These findings also open new questions for future research. One question is whether the microstructural changes observed here are linked to changes in visual function during infancy. Recent research suggests that there is some level of retinotopic functional connectivity in early infancy⁶⁰ and that by 5 months of age there is some level of retinotopic organization in early and intermediate human visual areas⁶¹ (V1, V2, V3, V3ab, V4). As neural receptive fields that underlie these maps develop during infancy^{62,63} and continue to develop during childhood^{64,65}, our data suggest the possibility that microstructural development of retinotopic visual areas found here will be coupled with fine tuning of receptive fields and retinotopic maps in the infant brain. Future research may determine if such

neural fine tuning is also associated with the rapid development of visual spatial acuity⁶⁶ and contrast sensitivity⁶⁷ during early infancy. ”

7. Authors identified genes related to postnatal development of sensory cortices. They then estimated the enriched physiological processes. From the next sub-chapter (“Early visual areas are more myelinated at birth but myelinate at slower than later visual areas”), author use relatively strong statement on myelin for interpreting MRI data. I think that there is some logical gap here. The enrichment analysis showed multiple physiological processes, including not only myelin-related process, but also synapse signaling, axonal projections and dendritic spines. I thought that there remains a possibility that processes other than myelination can also account for developmental change of qT1 and MD data. If authors have strong reasons that non-myelin factors identified by gene analysis will not explain variance of MD and qT1, please clarify. If authors consider that they could not fully exclude other factors, I will suggest toning down arguments on myelin in later part of Results and Discussion.

Response 20: We agree with the reviewer that although significant amount of the genes that were differentially expressed in the gene analysis were related to myelin and its processes, we also find significant evidence for the other biological mechanisms related to synaptogenesis and dendritic processes in the functional enrichment analysis. We also agree with the reviewer that as we are measuring quantitative T_1 or MD in cortex and not the white matter, there always remains a possibility that processes other than myelination also account for developmental change of qT1 and MD data (Gomez et al., 2017). Hence, we cannot not fully exclude other factors and we have now toned down the wording in the Results and Discussion as follows:

On page 11 of the Discussion, we write: *“Our data show strong effects of cortical increases in R_1 and decreases in MD supporting microstructural tissue growth in sensory-motor cortices from 0 to 6 months of age. These findings: (i) are supported by our transcriptomic analyses revealing myelination, synaptogenesis, and dendritic processes as key mechanisms of infants’ sensory-motor cortex development, and (ii) are consistent with prior histological data showing synaptogenesis¹¹ and dendritic growth⁶ in primate V1 and A1 during early infancy. While we do not find evidence for pruning, due to the complexity of factors that impact R_1 (T_1) in cortex, we acknowledge that we cannot conclude from qMRI metrics alone that there is no cortical pruning during early infancy. For example, it is possible that pruning effects may not be visible to qMRI metrics if the pruned neurites are unmyelinated. Additionally, changes in iron due to phagocytosis associated with pruning⁴³, may modulate R_1 (T_1), but may be obscured by larger effects of cortical myelination. Finally, our results do not preclude the possibility that pruning may occur later in infant development^{9,11} following this exuberant microstructural tissue growth during the first 6 months of life. However, our transcriptomic data provide complementary evidence for growth of multiple cortical tissue components during early infancy. Future histological investigations in pediatric samples containing sensory-motor cortices can elucidate precisely which cellular components develop during infancy in each visual area.”*

Specific points:

p.3: “E.g., at 3 months,” may be “For example, at 3 months,”

Response 21: On page 3 we have changed “E.g., at 3 months,” to “For example, at 3 months....”.

p.7: “mean diffusivity” has been already defined as MD in prior sections. I think that authors can simply use “MD”.

Response 22: On page 7, as the author noted as we have previously defined mean diffusivity in the manuscript, we changed mean diffusivity (MD) to MD.

p.9: Authors wrote “We find no evidence for cortical tissue pruning in sensory-motor cortices from 0 to 6 months of age”.

I’m not sure whether this argument can be supported by the data. A key assumption of the argument that

qT1 and/or MD is sensitive to identify cortical tissue pruning. If authors have confidence of this argument, please include explanations on why qT1 and MD must be sensitive for such tissue change.

Response 23: We agree with the reviewer that due to the complexity of factors that influence quantitative T_1 / R_1 metrics in cortex, it is indeed challenging to conclude from these metrics alone that there is no cortical tissue pruning. Hence, we now expand our discussion to address this point.

On page 11, we write: *“Our data show strong effects of cortical increases in R_1 and decreases in MD supporting microstructural tissue growth in sensory-motor cortices from 0 to 6 months of age. These findings: (i) are supported by our transcriptomic analyses revealing myelination, synaptogenesis, and dendritic processes as key mechanisms of infants’ sensory-motor cortex development, and (ii) are consistent with prior histological data showing synaptogenesis¹¹ and dendritic growth⁶ in primate V1 and A1 during early infancy. While we do not find evidence for pruning, due to the complexity of factors that impact R_1 (T_1) in cortex, we acknowledge that we cannot conclude from qMRI metrics alone that there is no cortical pruning during early infancy. For example, it is possible that pruning effects may not be visible to qMRI metrics if the pruned neurites are unmyelinated. Additionally, changes in iron due to phagocytosis associated with pruning⁴³, may modulate R_1 (T_1), but may be obscured by larger effects of cortical myelination. Finally, our results do not preclude the possibility that pruning may occur later in infant development^{9,11} following this exuberant microstructural tissue growth during the first 6 months of life. However, our transcriptomic data provide complementary evidence for growth of multiple cortical tissue components during early infancy. Future histological investigations in pediatric samples containing sensory-motor cortices can elucidate precisely which cellular components develop during infancy in each visual area.”*

p.10: Authors used two different magnets for acquiring the data. It will be helpful to cite some references showing that qT1 may be relatively stable across different magnets, to justify this data collection strategy.

Response 24: The reviewer asks us add references showing that qMRI metrics are reliable across magnets. We have added two citations that have shown that quantitative T_1/R_1 metrics are replicable across scanners. Specifically, these studies show that the discrepancies between qMRI data acquired with different scanner models are low if identical acquisition sequences and identical protocols are used for data collection, which is the case in our study. Consistent with this prior work, we also note in our own data the values across the subjects and timepoints are consistent across the two scanners.

On page 15 of the Methods, we write: *“Data were acquired at two identical 3T GE Discovery MR750 Scanners (GE Healthcare) and Nova 32-ch head coils (Nova Medical) located at Stanford University: (i) Center for Cognitive and Neurobiological Imaging (CNI) and (ii) Lucas Imaging Center using identical acquisition sequences and protocols. Prior work has shown that quantitative imaging metrics are replicable across scanners and subjects and have low inter-site discrepancies^{68,69}.”*

p.14: Some sentences of “dMRI” section misses period (“.”). Please also clarify whether authors used DTI or DKI to estimate MD.

Response 25: We thank the reviewer for pointing out the missing periods. We have added the missing periods in the dMRI paragraph. We also clarify that we used diffusional kurtosis files to estimate MD.

On page 16, we write: *“T2-weighted images were used for segmentation of gray-white matter to generate cortical surface reconstructions and dMRI data and diffusional kurtosis imaging was used to estimate MD in each voxel.”*

On page 18, we write: *“From the diffusional kurtosis tensor files, we estimated mean diffusivity (MD) maps in each voxel of the brain.”*

REVIEWERS' COMMENTS:

Reviewer #1 (Remarks to the Author):

The authors have produced a very detailed revision, diligently addressing the concerns raised by all reviewers of the manuscript. I don't have further comments and I would like to congratulate the authors on putting together a high-standard piece of work.

Reviewer #3 (Remarks to the Author):

In my opinion, authors have done an excellent job to address my comments. I have no further suggestions.